# Contractive Anchor Resolvent Diffusion for Incomplete Multi-View Clustering

**Tongzheng Zhao**[1]  **Yangyang Wen**[1]  **Yukai Shi**[1]  **Xinyan Liang**[1]  **Feijiang Li**[1]  **Peng Zhou**[2]  **Liang Du**[1]

## Abstract

Incomplete Multi-View Clustering (IMVC) is affected not only by missing feature values, but also by the degradation of relational structure induced by missing views. Many graph-based approaches either rely on costly data imputation or adopt first-order fusion mechanisms, which can be viewed as shallow low-pass filters with limited spectral selectivity. As a result, they may be insufficient to distinguish latent consensus structure from view-specific structural variations. To address this limitation, we reformulate IMVC from a spectral filtering perspective and propose **C**ontractive **A**nchor **R**esolvent **D**iffusion (**CARD**), a scalable framework for structural refinement without explicit view imputation. CARD constructs a unified anchor-induced hypergraph from observed sample–anchor relations and derives a high-order resolvent diffusion operator that acts as a rational spectral filter. This operator enhances the relative response of consensus-dominant modes while attenuating view-specific variations. We further derive a compact implicit solver that couples similarity learning and clustering without materializing dense matrices, and provide a conditional local refinement analysis under spectral-gap and local-stability assumptions. Extensive experiments on eight benchmarks, including large-scale datasets, show that CARD achieves competitive performance while scaling linearly in $N$ for a fixed anchor budget. The code for our method is publicly available at https://github.com/Whale-Waves/CARD.

## 1. Introduction

Multi-view clustering integrates heterogeneous information from multiple domains to uncover latent data structures. In real-world applications, data incompleteness is ubiquitous due to sensor failures, privacy restrictions, or asynchronous acquisition. Missing views do not only remove feature entries. They can also distort the relational structure across views, introduce noise into similarity graphs, and weaken the connectivity patterns required for accurate clustering.

Standard approaches to Incomplete Multi-View Clustering (IMVC) generally fall into two categories: imputation-based and fusion-based methods. Imputation-based methods (Lin et al., 2023; Xu et al., 2022; Zhang et al., 2025; Jiang et al., 2025b; Xu et al., 2024) employ generative models such as GANs or VAEs to synthesize missing data, but may incur substantial computational overhead and introduce synthetic artifacts that distort the original manifold. Fusion-based methods (Wen et al., 2021; 2023a; Liu et al., 2025; Jiang et al., 2025a; Guo et al., 2025; Dai et al., 2025) bypass explicit imputation and directly learn a consensus similarity graph from observed views, yet many of them rely on first-order mechanisms such as weighted averaging or co-regularization.

From a Graph Signal Processing (GSP) perspective, these first-order fusion operations can be interpreted as shallow low-pass filters. Their limited spectral selectivity makes it difficult to separate latent consensus structure from view-specific variations induced or amplified by missing views. This motivates a spectral filtering view of IMVC, where the goal is not to reconstruct missing features explicitly, but to refine the degraded relational structure under partial observations.

Motivated by this insight, we propose **C**ontractive **A**nchor **R**esolvent **D**iffusion (**CARD**), a scalable framework that reformulates IMVC as a high-order spectral filtering problem. CARD performs inference at the structural level by constructing a unified anchor-induced hypergraph from observed sample–anchor relations and employing a high-order resolvent operator to diffuse clustering-induced similarity over this relation graph. The resulting operator provides tunable spectral selectivity beyond standard first-order fusion. We further couple the resolvent diffusion with a clustering-induced similarity objective. For each fixed clustering assignment, the similarity update admits a closed-form resolvent solution; for each fixed similarity kernel, the assignment update is reduced to an exact Euclidean K-means

---

[1]Shanxi University, Taiyuan, China [2]Anhui University, Anhui, China. Correspondence to: Liang Du <duliang@sxu.edu.cn>.

*Proceedings of the 43rd International Conference on Machine Learning*, Seoul, South Korea. PMLR 306, 2026. Copyright 2026 by the author(s).

reformulation through a compact implicit factorization. Under spectral-gap and local-stability assumptions, we also provide a conditional local refinement analysis toward the consensus subspace.

Our main contributions are summarized as follows:

- **Spectral Reformulation and Modeling.** We view structural degradation as a key challenge in IMVC and formulate clustering-induced similarity learning as a high-order spectral filtering problem over observed sample–anchor relations. This leads to an imputation-free resolvent diffusion framework for refinement.

- **Scalable Implicit Optimization.** We derive a compact implicit solver for the high-order resolvent update. The solver avoids dense $N \times N$ matrix materialization and yields an exact Euclidean K-means reformulation for each fixed-kernel $\mathbf{Y}$-update, leading to linear scaling in $N$ for a fixed anchor budget.

- **Analysis and Empirical Validation.** We provide a spectral-filter interpretation, a positive-semidefinite compact factorization, and a conditional local refinement analysis under spectral-gap and local-stability assumptions. Experiments on eight benchmark datasets show that CARD achieves competitive or best performance in most tested settings, including large-scale incomplete scenarios.

## 2. Related Work

### 2.1. Incomplete Multi-view Clustering

The central challenge of Incomplete Multi-view Clustering (IMVC) is to infer a reliable consensus structure from partially observed multi-view data. Existing methods can be broadly grouped into two categories.

**Recovery-driven methods** explicitly estimate missing views or missing representations using generative or predictive mechanisms, such as cross-view completion (Lin et al., 2021), adaptive imputation (Jiang et al., 2025b), and prototype-guided synthesis (Yuan et al., 2024). Recent deep generative or diffusion-inspired models (Zhang et al., 2025) have also been explored to improve recovery quality. These methods can make use of cross-view correlations, but they may suffer from error propagation when the recovered views contain synthetic artifacts or outliers, as also discussed in Wang et al. (2024b). In addition, the computational cost of deep recovery modules can limit their applicability to large-scale datasets.

**Imputation-free methods** avoid explicit reconstruction and directly learn unified representations or similarity structures from observed entries (Xu et al., 2024; Wang et al., 2024a). This strategy reduces the risk of reconstruction artifacts.

However, similarity-based approaches (Liu et al., 2024d; Jiang et al., 2025a) often need to manipulate dense pairwise graphs, leading to an $\mathcal{O}(N^2)$ bottleneck. Moreover, many imputation-free methods rely on first-order fusion mechanisms, such as weighted averaging or co-regularization of view-specific graphs. Such operations provide useful local smoothing, but their limited spectral selectivity may restrict their ability to propagate structural information across weakly connected regions under high missing rates.

### 2.2. High-order Structure Modeling

Most graph-based IMVC methods build pairwise affinities, which are local and can be sensitive to missing or unreliable links (Guo et al., 2025; Jiang et al., 2025a). To capture more complex dependencies, tensor constraints, high-order proximity graphs, and hypergraph models have been studied (Yao et al., 2025). Hypergraphs are particularly suitable for representing group-wise relations, since anchors or prototypes can be viewed as hyperedges connecting multiple samples. Nevertheless, many existing high-order models use pre-defined or weakly adaptive topologies, and their role as spectral propagation operators under incomplete observations has been less explored. CARD follows this direction but defines a high-order resolvent diffusion operator on an anchor-induced relation graph, allowing clustering-induced similarity to be propagated beyond local pairwise neighborhoods without explicit view imputation.

### 2.3. Scalable Anchor-based Multi-view Clustering

Anchor-based multi-view clustering improves scalability by replacing dense sample–sample graphs with sample–anchor bipartite graphs (Qin et al., 2025a; Xu et al., 2025; Zhao et al., 2026a;b). With a fixed anchor budget, these methods can scale linearly with the number of samples. However, their performance can degrade in incomplete settings when anchors are shifted or semantically misaligned across views (Liu et al., 2024c; Wang et al., 2025b). Recent scalable methods such as RISE (Wang et al., 2025c) improve robustness, but their propagation mechanisms remain based on relatively shallow anchor-level structure learning. High-order frameworks such as AHMvC (Pan & Kang, 2023) mine richer interactions, yet they do not explicitly use a resolvent-type spectral mechanism to refine the degraded consensus structure induced by missing views. CARD bridges these aspects by combining scalable anchor relations with high-order resolvent diffusion.

### 2.4. Cross-view Relation Propagation

Cross-view relation modeling is important when the correspondence between views is noisy or incomplete. Recent studies have explored selective topology learning (Dong et al., 2025), robust contrastive alignment (Guo et al., 2024),

and flexible anchor learning (Qin et al., 2025b). These works suggest that the propagation topology across views is as important as the representation itself. CARD is motivated by this observation. Instead of relying only on direct identity-level coupling across views, CARD builds an anchor-induced diffusion topology over observed sample–anchor relations. This topology allows samples with incomplete observations to receive structural support from broader anchor-mediated neighborhoods, while the clustering-induced similarity and the diffusion operator are optimized jointly.

## 3. Methodology

In this section, we present the Contractive Anchor Resolvent Diffusion (CARD) framework. Unlike conventional incomplete multi-view clustering approaches that treat missing views mainly as partially observed feature vectors, CARD formulates the problem as learning from partial relational evidence. CARD performs high-order spectral diffusion over observed anchor-induced relations without explicitly imputing missing views. Under the standard IMVC assumption that each sample has at least one observed view, this construction can handle general missing-view patterns while retaining an anchor-based scalable representation.

### 3.1. Problem Formulation and Preliminaries

Let $\mathcal{X} = \{\mathbf{x}_i^{(v)} : i = 1, \ldots, N, \ v = 1, \ldots, V\}$ denote a multi-view dataset with $N$ samples and $V$ heterogeneous views, where $\mathbf{x}_i^{(v)}$ is available only when sample $i$ is observed in view $v$. View availability is encoded by $\mathbf{\Pi} \in \{0, 1\}^{N \times V}$, where $\Pi_{iv} = 1$ indicates that sample $i$ is observed in view $v$, and $\Pi_{iv} = 0$ otherwise. We follow the standard IMVC setting and assume that each sample has at least one observed view, i.e., $\sum_{v=1}^{V} \Pi_{iv} > 0$ for $i = 1, \ldots, N$.

Our objective is to jointly learn a unified similarity matrix $\mathbf{C} \in \mathbb{R}^{N \times N}$ and a cluster assignment matrix $\mathbf{Y} \in \{0, 1\}^{N \times c}$ from observed multi-view relations and clustering-induced semantic consistency. Relational evidence is encoded by availability-gated sample–anchor associations. Thus, missing sample–view pairs contribute no relational edges to the diffusion operator, rather than being filled with artificial feature values or synthetic similarities.

### 3.2. Anchor-Induced Relational Construction

To avoid constructing dense and incomplete sample-to-sample graphs, we represent observed relations through sparse sample–anchor associations. For each view $v$, let $\mathcal{I}_v = \{i : \Pi_{iv} = 1\}$ be the index set of observed samples and $n_v = |\mathcal{I}_v|$. We select $m_v$ anchors from the observed samples in view $v$ using K-means and construct a nonnegative bipartite affinity matrix $\mathbf{B}^{(v)} \in \mathbb{R}^{n_v \times m_v}$ on the observed subset.

For an observed sample $i \in \mathcal{I}_v$, its affinities to the anchors are normalized as $H_{ij}^{(v)} = B_{ij}^{(v)}/(\sum_{k=1}^{m_v} B_{ik}^{(v)} + \epsilon)$, $j = 1, \ldots, m_v$, where $\epsilon > 0$ is a small constant for numerical stability. In practice, each observed sample is connected only to its $\delta$ nearest anchors, which makes the sample–anchor incidence sparse. We then embed the view-specific incidence matrix into an $N \times m_v$ matrix $\hat{\mathbf{H}}^{(v)}$ by assigning zero rows to unobserved samples: $\hat{H}_{ij}^{(v)} = H_{ij}^{(v)}$ if $\Pi_{iv} = 1$, and $\hat{H}_{ij}^{(v)} = 0$ otherwise. Equivalently, unobserved sample–view pairs contribute no edges to the anchor-induced graph.

We concatenate all view-specific gated incidence matrices as $\tilde{\mathbf{H}} = [\hat{\mathbf{H}}^{(1)}, \ldots, \hat{\mathbf{H}}^{(V)}] \in \mathbb{R}^{N \times M}$, where $M = \sum_{v=1}^{V} m_v$. Let $\mathbf{W} \in \mathbb{R}^{M \times M}$ denote a diagonal anchor weight matrix with positive diagonal entries. The vertex and anchor degree matrices are defined as $D_{ii} = \sum_{k=1}^{M} \tilde{H}_{ik} W_{kk}$ and $(\mathbf{D}_e)_{kk} = \sum_{i=1}^{N} \tilde{H}_{ik}$, respectively. Zero-degree anchors, if any, are removed before normalization. Under the above view-availability assumption, the constructed incidence graph assigns each sample at least one observed anchor connection, so $D_{ii} > 0$ for all samples participating in clustering.

### 3.3. Anchor-Hypergraph Diffusion Operator

Based on the unified anchor-induced hypergraph, we define the normalized bipartite diffusion operator

$$\mathbf{Q} = \mathbf{D}^{-1/2}\tilde{\mathbf{H}}\mathbf{W}^{1/2}\mathbf{D}_e^{-1/2}. \qquad (1)$$

The induced sample-to-sample operator is $\mathbf{A} = \mathbf{Q}\mathbf{Q}^\top$. Since $\mathbf{A}$ is symmetric and positive semi-definite, it serves as an anchor-induced smoothing operator over the observed relational structure. We use the normalized operator, or equivalently rescale it when necessary, so that $\|\mathbf{A}\|_2 \leq 1$, which ensures that the resolvent operator used below is well-defined for $\alpha \in (0, 1)$.

The diffusion operator only uses paths supported by observed sample–anchor evidence. Unobserved sample–view pairs contribute zero rows to the corresponding view-specific incidence matrix and therefore introduce no artificial edges into $\tilde{\mathbf{H}}$. Nevertheless, a sample with missing views can still receive structural support through its observed views and their anchor-mediated neighborhoods. Thus, CARD propagates information through the observed anchor skeleton rather than through imputed features or synthetic similarities.

Under partial observations, the diffusion regularization encourages high-order relational consistency. Intuitively, two similarity entries $C_{ij}$ and $C_{i'j'}$ are encouraged to be close when the corresponding sample pairs are connected through similar anchor-induced relational patterns. If a sample–view relation is unobserved, the corresponding incidence weight

is zero and does not contribute to the diffusion regularizer. This intuition is formalized by the high-order consistency energy discussed in Appendix A.1, which motivates the normalized diffusion regularizer in Eq. (2).

### 3.4. High-Order Consensus Optimization

CARD jointly optimizes a similarity matrix $\mathbf{C}$ and a cluster assignment matrix $\mathbf{Y}$ by coupling anchor-induced structural diffusion with clustering-induced semantic consistency. Instead of using first-order graph fusion, we employ a high-order diffusion regularizer that favors similarity matrices stable under anchor-induced smoothing. For a fixed clustering assignment $\mathbf{Y}$, the similarity update admits a closed-form resolvent solution obtained from a regularized high-order Dirichlet energy. For a fixed similarity kernel, the assignment update is exactly equivalent to Euclidean K-means in a dynamically constructed feature space; in practice, we optimize this equivalent K-means problem using standard Lloyd iterations.

The optimization problem is formulated as

$$\min_{\mathbf{C},\mathbf{Y}} \quad \mathrm{Tr}\big(\mathbf{C}^\top\mathbf{C} - \mathbf{C}^\top\mathbf{A}\mathbf{C}\mathbf{A}^\top\big) + \mu\|\mathbf{C} - \mathbf{S}(\mathbf{Y})\|_F^2 \quad (2)$$

$$\text{s.t.} \quad \mathbf{Y} \in \{0,1\}^{N\times c}, \qquad \mathbf{Y}\mathbf{1}_c = \mathbf{1}_N.$$

Here, $\mathbf{S}(\mathbf{Y}) = \mathbf{Y}(\mathbf{Y}^\top\mathbf{Y})^{-1}\mathbf{Y}^\top$ is the clustering-induced similarity matrix. Its entry $S_{ij}(\mathbf{Y})$ is positive only when samples $i$ and $j$ are assigned to the same cluster, with normalization by the corresponding cluster size. We consider assignments with nonempty clusters so that $\mathbf{Y}^\top\mathbf{Y}$ is invertible. The similarity matrix $\mathbf{C}$ is optimized over the unconstrained real matrix space. For fixed $\mathbf{Y}$, both $\mathbf{A}$ and $\mathbf{S}(\mathbf{Y})$ are symmetric. Since $\mu > 0$ and $\|\mathbf{A}\|_2 \leq 1$, the fixed-$\mathbf{Y}$ subproblem has a unique minimizer, and the resulting optimal solution is symmetric. Therefore, symmetry of $\mathbf{C}$ does not need to be imposed as an explicit constraint.

### 3.5. Optimization

The objective in Eq. (2) couples the continuous similarity matrix $\mathbf{C}$ and the discrete cluster assignment matrix $\mathbf{Y}$. Directly optimizing $\mathbf{C}$ in the full $N \times N$ space is computationally prohibitive, and a naive kernel K-means update would require materializing a dense kernel matrix. We show that the fixed-$\mathbf{Y}$ similarity update admits a compact closed-form solution, and that the fixed-kernel $\mathbf{Y}$-subproblem is exactly equivalent to Euclidean K-means in an explicit feature space.

#### 3.5.1. CLOSED-FORM SPECTRAL RESOLUTION

For a fixed cluster assignment $\mathbf{Y}$, the optimization with respect to $\mathbf{C}$ reduces to

$$\min_{\mathbf{C}} \quad \mathrm{Tr}\big(\mathbf{C}^\top\mathbf{C} - \mathbf{C}^\top\mathbf{A}\mathbf{C}\mathbf{A}^\top\big) + \mu\|\mathbf{C} - \mathbf{S}\|_F^2, \quad (3)$$

where $\mathbf{S} = \mathbf{S}(\mathbf{Y})$. Using the symmetry of $\mathbf{A}$, the stationary condition with respect to $\mathbf{C}$ gives

$$(1 + \mu)\mathbf{C} - \mathbf{A}\mathbf{C}\mathbf{A}^\top = \mu\mathbf{S}. \quad (4)$$

Since $\mathbf{A} = \mathbf{Q}\mathbf{Q}^\top$ is induced by the compact anchor representation, Eq. (4) can be solved in the anchor-induced spectral subspace rather than by a dense Sylvester solver. With $\alpha = 1/(1+\mu) \in (0,1)$, Eq. (4) is equivalently written as

$$\mathbf{C} - \alpha\mathbf{A}\mathbf{C}\mathbf{A}^\top = (1-\alpha)\mathbf{S}, \quad (5)$$

which corresponds to a resolvent operator associated with $\mathbf{A} \otimes \mathbf{A}$. The parameter mapping and operator interpretation are detailed in Appendix A.2.

**Theorem 3.1** (Compact Spectral Filtering and Factorization). *Assume that the normalized anchor-induced operator satisfies $\mathbf{A} = \mathbf{Q}\mathbf{Q}^\top \succeq 0$ and $\|\mathbf{A}\|_2 \leq 1$. Let $r = \mathrm{rank}(\mathbf{Q}) = \mathrm{rank}(\mathbf{A}) \leq M$, and consider the compact SVD*

$$\mathbf{Q} = \mathbf{U}_r\boldsymbol{\Sigma}_r\mathbf{V}_r^\top. \quad (6)$$

*It follows that*

$$\mathbf{A} = \mathbf{Q}\mathbf{Q}^\top = \mathbf{U}_r\boldsymbol{\Sigma}_r^2\mathbf{U}_r^\top. \quad (7)$$

*Thus, the nonzero eigenvalues of $\mathbf{A}$ are $\lambda_i = \sigma_i^2$, and $0 \leq \lambda_i = \sigma_i^2 \leq 1$. The fixed-$\mathbf{Y}$ optimal solution of Eq. (4) admits the compact representation*

$$\mathbf{C}^* = (1-\alpha)\mathbf{S} + \mathbf{U}_r\left(\boldsymbol{\Omega}_r \circ (\mathbf{U}_r^\top\mathbf{S}\mathbf{U}_r)\right)\mathbf{U}_r^\top, \quad (8)$$

*where $\circ$ denotes the Hadamard product, and $\boldsymbol{\Omega}_r \in \mathbb{R}^{r\times r}$ is defined element-wise as*

$$[\boldsymbol{\Omega}_r]_{ij} = \frac{\alpha(1-\alpha)\sigma_i^2\sigma_j^2}{1 - \alpha\sigma_i^2\sigma_j^2}, \qquad 1 \leq i,j \leq r. \quad (9)$$

*Moreover, under the same normalization, $\boldsymbol{\Omega}_r$ is positive semi-definite for any $\alpha \in (0,1)$. Hence there exists a factorization $\boldsymbol{\Omega}_r = \mathbf{G}_r\mathbf{G}_r^\top$, where $\mathbf{G}_r \in \mathbb{R}^{r\times r}$. The derivation of Eq. (8) and the PSD proof are given in Appendix A.3 and Appendix A.4, respectively.*

Theorem 3.1 gives a compact spectral-filter interpretation of the fixed-$\mathbf{Y}$ update. Components associated with larger anchor-induced singular values receive a stronger relative response through the resolvent factor, while components outside the anchor subspace retain the residual weight $(1 - \alpha)\mathbf{S}$. This compact form also enables the explicit feature construction used in the assignment update.

#### 3.5.2. DYNAMIC EXPLICIT FEATURE MAPPING

We now consider the subproblem of updating the cluster assignment with the current kernel $\mathbf{C}^{(t)}$ fixed. Let $\tilde{\mathbf{Y}} = \mathbf{Y}(\mathbf{Y}^\top\mathbf{Y})^{-1/2}$ be the normalized indicator matrix. Since

$\|\mathbf{S}(\mathbf{Y})\|_F^2 = c$ for nonempty clusters, minimizing $\|\mathbf{C}^{(t)} - \mathbf{S}(\mathbf{Y})\|_F^2$ with respect to $\mathbf{Y}$ is equivalent to maximizing the kernel K-means objective

$$\max_{\mathbf{Y}} \mathcal{J}_{\mathbf{Y}}^{(t)} = \mathrm{Tr}\left(\tilde{\mathbf{Y}}^\top \mathbf{C}^{(t)} \tilde{\mathbf{Y}}\right) = \sum_{k=1}^{c} \frac{\mathbf{y}_k^\top \mathbf{C}^{(t)} \mathbf{y}_k}{\mathbf{y}_k^\top \mathbf{y}_k}, \quad (10)$$

where $\mathbf{y}_k \in \{0, 1\}^N$ is the indicator vector of cluster $k$. A direct implementation of Eq. (10) would require the dense kernel $\mathbf{C}^{(t)}$, incurring $\mathcal{O}(N^2)$ memory.

The following result shows that this fixed-kernel subproblem can be represented exactly as Euclidean K-means on a dynamically constructed feature embedding.

**Theorem 3.2** (Iterative Kernel-to-Euclidean Reformulation). *Let $\mathbf{Q} = \mathbf{U}_r \boldsymbol{\Sigma}_r \mathbf{V}_r^\top$ be the compact SVD of $\mathbf{Q}$. At iteration $t$, let $\mathbf{Y}^{(t)}$ be the current assignment, and let $\mathbf{C}^{(t)}$ be the similarity matrix computed from $\mathbf{Y}^{(t)}$ by Theorem 3.1. Define $\tilde{\mathbf{Y}}^{(t)} = \mathbf{Y}^{(t)}((\mathbf{Y}^{(t)})^\top \mathbf{Y}^{(t)})^{-1/2}$ and $\mathbf{J}^{(t)} = \mathbf{U}_r^\top \tilde{\mathbf{Y}}^{(t)}$. Let $\boldsymbol{\Omega}_r = \mathbf{G}_r \mathbf{G}_r^\top$, and denote the columns of $\mathbf{G}_r$ by $\{\mathbf{g}_i\}_{i=1}^r$. Define*

$$\mathbf{P}^{(t)} = \left[\mathrm{diag}(\mathbf{g}_1)\mathbf{J}^{(t)}, \ldots, \mathrm{diag}(\mathbf{g}_r)\mathbf{J}^{(t)}\right] \in \mathbb{R}^{r \times rc}. \quad (11)$$

*Then $\mathbf{C}^{(t)}$ admits the exact factorization $\mathbf{C}^{(t)} = \mathbf{Z}^{(t)}(\mathbf{Z}^{(t)})^\top$, where*

$$\mathbf{Z}^{(t)} = \left[\mathbf{U}_r \mathbf{P}^{(t)}, \sqrt{1-\alpha}\, \tilde{\mathbf{Y}}^{(t)}\right] \in \mathbb{R}^{N \times (rc+c)}. \quad (12)$$

*Consequently, the fixed-kernel objective in Eq. (10) is equivalent to minimizing the Euclidean K-means objective on $\mathbf{Z}^{(t)}$, up to a fixed constant.*

Theorem 3.2 establishes an exact reformulation of the fixed-kernel K-means subproblem. The iteration uses the current assignment $\mathbf{Y}^{(t)}$ to construct $\mathbf{C}^{(t)}$ and its explicit factor $\mathbf{Z}^{(t)}$, and then updates the assignment by applying Euclidean K-means to $\mathbf{Z}^{(t)}$. The equivalence is exact at the subproblem level; in practice, this step is optimized by Lloyd iterations and is therefore subject to the usual local-optimum behavior of K-means.

### 3.5.3. ALGORITHM AND COMPLEXITY

Guided by Theorem 3.2, CARD alternates between **feature construction** and **discrete clustering**. In the feature construction step, the current assignment $\mathbf{Y}^{(t)}$ is used to form the embedding $\mathbf{Z}^{(t)}$ in Eq. (12), combining the static anchor topology with the current clustering structure without explicitly forming $\mathbf{C}^{(t)}$. In the clustering step, Euclidean K-means is applied to $\mathbf{Z}^{(t)}$ to obtain the next assignment. The full procedure is summarized in Algorithm 1 in Appendix A.6.

The initialization computes the compact spectral decomposition through the anchor side and costs $\mathcal{O}(NM^2 + M^3)$

in the dense anchor representation. With sparse top-$\delta$ anchor connections, operations involving $\mathbf{Q}$ are implemented through sparse sample–anchor multiplications. During refinement, the dominant operations are constructing $\mathbf{Z}^{(t)}$ and running K-means on an $N \times (rc + c)$ embedding, giving the per-iteration cost

$$\mathcal{O}\left(NV\delta rc + Mr^2c + T_{\mathrm{km}}Nrc^2\right), \quad (13)$$

where $T_{\mathrm{km}}$ is the number of Lloyd iterations. For a fixed anchor budget, fixed $\delta$, and fixed number of clusters $c$, the cost scales linearly with $N$, and no dense $N \times N$ matrix is materialized. With sparse top-$\delta$ anchor connections, the memory footprint is $\mathcal{O}(NV\delta + Nrc + Mr)$, excluding raw feature storage.

## 4. Theoretical Analysis

This section provides theoretical support for the main components of CARD. We focus on properties that directly follow from the proposed resolvent operator: spectral selectivity, positive path-supported diffusion, conditioning, and first-order stability to anchor perturbations. The analysis is not intended as a global recovery or global convergence guarantee for the full alternating discrete optimization problem.

### 4.1. Operator-Theoretic Setup

We base the analysis on the unified anchor-induced operator $\mathbf{A}$ defined in Section 3. Let $\mathbf{Q} = \mathbf{U}_r \boldsymbol{\Sigma}_r \mathbf{V}_r^\top$ be the compact SVD of $\mathbf{Q}$. Then $\mathbf{A} = \mathbf{Q}\mathbf{Q}^\top = \mathbf{U}_r \boldsymbol{\Sigma}_r^2 \mathbf{U}_r^\top$ is the induced spectral factorization of $\mathbf{A}$, where $r = \mathrm{rank}(\mathbf{Q}) = \mathrm{rank}(\mathbf{A})$. Equivalently, if $\lambda_i$ denotes the $i$-th nonzero eigenvalue of $\mathbf{A}$, then $\lambda_i = \sigma_i^2$. Throughout this section, we assume that $\mathbf{A} \succeq 0$ and $\|\mathbf{A}\|_2 \leq 1$. This condition is ensured by the normalized construction in Section 3, or by an equivalent rescaling when necessary.

For spectral interpretation, we consider a separated regime in which the leading modes of $\mathbf{A}$ correspond to consensus structure and the remaining active modes correspond to residual variations. Specifically, let $1 \geq \lambda_{\mathrm{sig}} > \lambda_{\mathrm{var}} \geq 0$ denote representative signal and variation eigenvalues in the active anchor subspace. This separation assumption is used only to interpret the spectral response of the resolvent filter.

**Definition 4.1** (High-Order Resolvent Operator). For $\alpha \in (0, 1)$, the high-order resolvent operator $\mathcal{R}_\alpha$ is defined by

$$\mathrm{vec}(\mathbf{C}) = \mathcal{R}_\alpha(\mathbf{S}) \triangleq (1 - \alpha)\left(\mathbf{I} - \alpha\mathbf{A} \otimes \mathbf{A}\right)^{-1} \mathrm{vec}(\mathbf{S}). \quad (14)$$

Since $\|\mathbf{A}\|_2 \leq 1$ and $\alpha \in (0, 1)$, the inverse in Eq. (14) is well-defined. The operator $\mathcal{R}_\alpha$ is exactly the fixed-$\mathbf{Y}$ stationary solution of the high-order diffusion objective in Eq. (2). The mapping between $\mu$ and $\alpha$ is given in Appendix A.2.

## 4.2. Spectral Selectivity of the Resolvent Filter

We first explain how the resolvent operator provides tunable spectral selectivity beyond shallow first-order smoothing. For an eigenmode with eigenvalue $\lambda$ of $\mathbf{A}$, the corresponding diagonal Kronecker mode has response

$$g_\alpha(\lambda) = \frac{1 - \alpha}{1 - \alpha\lambda^2}. \tag{15}$$

**Proposition 4.2** (Spectral Selectivity of the Resolvent Filter). *Under $1 \geq \lambda_{\text{sig}} > \lambda_{\text{var}} \geq 0$, define the spectral selectivity ratio as $\text{SSR}(\mathcal{R}_\alpha) = g_\alpha(\lambda_{\text{sig}})/g_\alpha(\lambda_{\text{var}})$. Then*

$$\text{SSR}(\mathcal{R}_\alpha) = \frac{1 - \alpha\lambda_{\text{var}}^2}{1 - \alpha\lambda_{\text{sig}}^2} > 1. \tag{16}$$

*Moreover, $\text{SSR}(\mathcal{R}_\alpha)$ is non-decreasing in $\alpha$. If $\lambda_{\text{sig}} = 1$, it scales as $\Theta((1 - \alpha)^{-1})$; otherwise it converges to $(1 - \lambda_{\text{var}}^2)/(1 - \lambda_{\text{sig}}^2)$ as $\alpha \to 1^-$.*

The derivation follows from Eq. (15) and is provided in Appendix B.1. Proposition 4.2 gives a spectral interpretation of CARD: when consensus modes have larger eigenvalues than residual variation modes, increasing $\alpha$ amplifies their relative response. This explains the filtering effect of the resolvent operator, without claiming a label-level recovery guarantee.

## 4.3. Positive Path-Supported Diffusion Contribution

The resolvent operator can assign positive structural contribution through multi-hop anchor-supported paths. The following result states a positivity property of the fixed-$\mathbf{S}$ diffusion map.

**Proposition 4.3** (Path-Supported Positive Diffusion Contribution). *Assume that $\mathbf{A}$ and $\mathbf{S}$ are entrywise nonnegative, and that $\|\mathbf{A}\|_2 \leq 1$. If there exist samples $p, q$, a path length $\ell \geq 1$, and constants $\eta_i, \eta_j, s_{\min} > 0$ such that*

$$(\mathbf{A}^\ell)_{ip} \geq \eta_i, \qquad (\mathbf{A}^\ell)_{jq} \geq \eta_j, \qquad S_{pq} \geq s_{\min}, \tag{17}$$

*then the resolvent solution satisfies*

$$C_{ij} \geq (1 - \alpha)\alpha^\ell \eta_i \eta_j s_{\min} > 0. \tag{18}$$

The proof is based on the Neumann-series expansion of $\mathcal{R}_\alpha$ and is given in Appendix B.2. This result should be interpreted as a positive path-contribution property of the diffusion operator: anchor-supported paths can contribute positive structural mass without assigning artificial feature values to missing views. It is not an explicit imputation or recovery guarantee.

## 4.4. Conditional Local Refinement Interpretation

We next state a local, assumption-driven refinement result. This result is intended to clarify how spectral selectivity may improve a nearby clustering subspace; it is not a global convergence theorem for the alternating algorithm.

**Proposition 4.4** (Conditional Local Subspace Refinement). *Let $\mathcal{V}^{(t)}$ be the subspace induced by $\mathbf{Y}^{(t)}$, and let $\mathcal{V}^*$ denote the consensus subspace. Define $d(\mathcal{V}, \mathcal{V}^*) = \|\sin\Theta(\mathcal{V}, \mathcal{V}^*)\|_F$. Assume that the current partition lies in a local basin of $\mathcal{V}^*$, the filtered kernel has a non-vanishing eigengap, and the continuous resolvent update satisfies a local subspace contraction of the form*

$$d_{\text{cont}}^{(t+1)} \leq C_{\text{loc}}\nu(\alpha)d(\mathcal{V}^{(t)}, \mathcal{V}^*), \tag{19}$$

*where $C_{\text{loc}} > 0$ is a local perturbation constant and*

$$\nu(\alpha) = \frac{1 - \alpha\lambda_{\text{sig}}^2}{1 - \alpha\lambda_{\text{var}}^2} \in (0, 1). \tag{20}$$

*If the subsequent K-means discretization introduces a bounded subspace error $\xi_{\text{disc}}$, then*

$$d(\mathcal{V}^{(t+1)}, \mathcal{V}^*) \leq C_{\text{loc}}\nu(\alpha)d(\mathcal{V}^{(t)}, \mathcal{V}^*) + \xi_{\text{disc}}. \tag{21}$$

*In particular, when $C_{\text{loc}}\nu(\alpha) < 1$, the recursion is locally contractive up to the discretization error.*

The proof is provided in Appendix B.3. Proposition 4.4 makes explicit the assumptions required for a contraction-style interpretation: a local basin, a filtered eigengap, and bounded discretization error. The factor $\nu(\alpha) < 1$ follows from $\lambda_{\text{sig}} > \lambda_{\text{var}}$, while the additional constant $C_{\text{loc}}$ accounts for the local perturbation geometry and prevents the statement from being read as an unconditional Davis–Kahan consequence.

## 4.5. Optimization Stability

Let $\mathcal{L} \triangleq \mathbf{I} - \alpha(\mathbf{A} \otimes \mathbf{A})$ be the linear operator associated with the resolvent system. Its conditioning governs the numerical stability of the implicit similarity update.

**Proposition 4.5** (Stability–Selectivity Trade-off). *Let $\lambda_1 = \|\mathbf{A}\|_2 \leq 1$. For $\mathcal{L} = \mathbf{I} - \alpha(\mathbf{A} \otimes \mathbf{A})$,*

$$\kappa_2(\mathcal{L}) \leq \frac{1}{1 - \alpha\lambda_1^2} \leq \frac{1}{1 - \alpha}. \tag{22}$$

*If $\mathbf{A}$ has a zero eigenvalue, the first inequality becomes equality. When $\lambda_1 \approx 1$, the upper bound scales as $\mathcal{O}((1 - \alpha)^{-1})$.*

The proof follows from the spectrum of $\mathbf{A} \otimes \mathbf{A}$ and is deferred to Appendix B.4. Together with Proposition 4.2, this result describes a trade-off: increasing $\alpha$ improves spectral selectivity, but may worsen the conditioning of the implicit linear system.

## 4.6. Robustness to Anchor Shift

In practice, anchors are estimated from data and may deviate from ideal landmarks. We analyze the first-order stability of the fixed-$\mathbf{S}$ resolvent map with respect to perturbations in the anchor embedding $\mathbf{Q}$.

**Proposition 4.6** (First-Order Stability to Anchor Perturbations). *Assume* $\|\mathbf{A}\|_2 \leq 1$, $\|\hat{\mathbf{A}}\|_2 \leq 1$, *and* $\|\hat{\mathbf{Q}} - \mathbf{Q}\|_2 \leq \epsilon$. *Let* $\mathbf{C}^*$ *and* $\hat{\mathbf{C}}$ *be the fixed-$\mathbf{S}$ resolvent solutions induced by* $\mathbf{Q}$ *and* $\hat{\mathbf{Q}}$, *respectively. Then, for sufficiently small* $\epsilon$, *up to first order,*

$$\|\hat{\mathbf{C}} - \mathbf{C}^*\|_F \leq \frac{4\alpha\|\mathbf{Q}\|_2^3}{1-\alpha}\epsilon\|\mathbf{S}\|_F + \mathcal{O}(\epsilon^2). \quad (23)$$

*In particular, if* $\|\mathbf{Q}\|_2 \leq 1$, *the leading term is bounded by* $4\alpha\epsilon\|\mathbf{S}\|_F/(1-\alpha)$.

The proof traces how perturbations in $\mathbf{Q}$ affect $\mathbf{A} = \mathbf{Q}\mathbf{Q}^\top$ and the resolvent system; see Appendix B.5. Proposition 4.6 gives a local first-order stability statement for the fixed-$\mathbf{S}$ similarity update. It does not cover discontinuities introduced by the subsequent discrete K-means assignment step.

# 5. Experiments

## 5.1. Experimental Settings

The detailed descriptions of the datasets (Appendix C.1) and the compared methods (Appendix C.2) are provided in the appendix. We report clustering accuracy (ACC) (Kuhn, 1955), normalized mutual information (NMI) (Strehl & Ghosh, 2002), and adjusted Rand index (ARI) (Hubert & Arabie, 1985).

**Incomplete settings.** For each dataset, we generate incomplete versions by randomly removing views with missing ratio $\rho = (n - n_p)/n \in \{0.1, 0.3, 0.5, 0.7, 0.9\}$, following IMVC-CBG (Wang et al., 2022b), where $n$ is the total number of samples and $n_p$ is the number of paired, i.e., fully observed, samples. For each dataset and each missing ratio, we generate 10 independent incomplete splits, run all methods on the same predefined splits, and report the average performance over the 10 runs.

**Complete settings.** We also evaluate all methods on fully observed data ($\rho = 0$), where the number of clusters $c$ is set to the ground-truth class number.

**Parameter selection protocol.** For a fair comparison, all methods, including CARD and the baseline methods, are evaluated under the same search-and-report protocol. For each baseline, we use the official implementation when available and search its parameters over the recommended ranges provided by the original paper or code release. For CARD, we search $\alpha \in \{0.1, 0.3, 0.5, 0.7, 0.9\}$, the anchor neighborhood size $\delta \in \{5, 10\}$, and the number of anchors

per view $m_v \in \{5c, 10c, 20c, 50c, 100c\}$. For each method and each evaluation metric, we report the best value over its corresponding search grid, following common practice in unsupervised clustering benchmarks. The same search-and-report protocol is applied to CARD and all baselines.

Here $m_v$ is specified *per view*; for example, when $m_v = 5c$ and the dataset has $V = 3$ views, the fused hypergraph contains $M = 15c$ anchors. In implementation, the effective anchor number is $m_v^{\text{eff}} = \min(m_v, n_v)$, where $n_v$ is the number of observed samples in view $v$ under the current split.

## 5.2. Results under Incomplete Settings

We evaluate CARD under incomplete settings with varying missing rates. To highlight robustness and scalability under severe missing-view scenarios, the main text reports datasets across small-, medium-, and large-scale settings, while results on additional small- and medium-scale datasets are provided in Table 4 of the appendix. Table 1 summarizes the results on these datasets.

Overall, CARD achieves the best or competitive results in most reported dataset-metric settings across different missing rates. This overall advantage is evident across small- and medium-scale datasets (UCIDigits and Scene-15) and large-scale datasets (CCV, SUNRGBD, and NUSWIDEOBJ), suggesting robustness under the tested severe missingness settings.

In terms of scalability, CARD exhibits clear practical advantages. On NUSWIDEOBJ, several competing methods fail due to out-of-memory issues, whereas CARD successfully completes all missing-rate settings and achieves the best or competitive results in most reported settings among runnable baselines. These results suggest that CARD is practically robust and scalable in the evaluated large-scale incomplete settings.

## 5.3. Results under Complete Settings

We further evaluate CARD under the complete setting ($\rho = 0$) to verify its effectiveness when all views are fully observed. As summarized in Table 5, CARD achieves the best or competitive performance on most datasets and metrics, indicating that the proposed approach remains effective when all views are fully observed.

## 5.4. Runtime Analysis

Computationally, CARD scales linearly with the number of samples for a fixed anchor budget. As confirmed by the runtime evaluation in Fig. 2, CARD scales to large datasets efficiently, is substantially faster than dense graph-based alternatives, and remains competitive with anchor-based

*Table 1.* Clustering performance on small-, medium-, and large-scale datasets under incomplete settings (missing rates $\rho = 0.1 - 0.9$). Datasets are sorted by sample size. "–" indicates that the method was not runnable under the same computational budget due to memory or time limitations.

| Dataset | Method | ACC | | | | | NMI | | | | | ARI | | | | |
|---|---|---|---|---|---|---|---|---|---|---|---|---|---|---|---|---|
| | | 0.1 | 0.3 | 0.5 | 0.7 | 0.9 | 0.1 | 0.3 | 0.5 | 0.7 | 0.9 | 0.1 | 0.3 | 0.5 | 0.7 | 0.9 |
| UCIDigits | PIC [IJCAI 2019] | 0.8625 | 0.8440 | 0.8660 | 0.8675 | 0.8570 | 0.8555 | 0.8319 | 0.8563 | 0.8603 | 0.8495 | 0.8080 | 0.7717 | 0.8125 | 0.8141 | 0.8133 |
| | EEIMVC [TPAMI 2021] | 0.9430 | 0.9330 | 0.8995 | 0.9020 | 0.8675 | 0.8810 | 0.8628 | 0.8007 | 0.8028 | 0.7517 | 0.8788 | 0.8578 | 0.7921 | 0.7949 | 0.7329 |
| | IMVC-CBG [CVPR 2022] | 0.8885 | 0.9190 | 0.8905 | 0.8855 | 0.8855 | 0.8070 | 0.8400 | 0.7919 | 0.7872 | 0.7848 | 0.7733 | 0.8309 | 0.7740 | 0.7675 | 0.7635 |
| | SIMVC-SA [MM 2023] | 0.8780 | 0.8890 | 0.8130 | 0.8200 | 0.8315 | 0.7689 | 0.7936 | 0.6905 | 0.6828 | 0.6982 | 0.7510 | 0.7736 | 0.6438 | 0.6472 | 0.6687 |
| | FCMVC-IVA [TIP 2024] | 0.7515 | 0.6735 | 0.6230 | 0.4960 | 0.3610 | 0.6254 | 0.5238 | 0.4521 | 0.3530 | 0.2800 | 0.5610 | 0.4452 | 0.3601 | 0.2389 | 0.1559 |
| | FIMVC-VIA [TNNLS 2024] | 0.9325 | 0.9230 | 0.9115 | 0.8925 | 0.8985 | 0.8591 | 0.8455 | 0.8219 | 0.7899 | 0.8025 | 0.8572 | 0.8392 | 0.8147 | 0.7789 | 0.7904 |
| | FSIMVC-OF [MM 2024] | 0.9530 | 0.9450 | 0.9330 | 0.9305 | 0.9165 | 0.8999 | 0.8944 | 0.8666 | 0.8573 | 0.8396 | 0.8990 | 0.8875 | 0.8603 | 0.8531 | 0.8287 |
| | LFIMVC [TIP 2024] | 0.8805 | 0.8535 | 0.8310 | 0.7885 | 0.8200 | 0.7876 | 0.7518 | 0.7280 | 0.6769 | 0.6537 | 0.7589 | 0.7180 | 0.6820 | 0.6252 | 0.6537 |
| | PSIMVC-PG [TNNLS 2024] | 0.8335 | 0.8950 | 0.8445 | 0.8435 | 0.7760 | 0.7810 | 0.8056 | 0.7466 | 0.7406 | 0.6779 | 0.7199 | 0.7849 | 0.7100 | 0.6981 | 0.6184 |
| | CAL [IJCAI 2025] | 0.8615 | 0.7805 | 0.6825 | 0.5965 | 0.5975 | 0.8850 | 0.7883 | 0.6572 | 0.5698 | 0.5622 | 0.8309 | 0.6664 | 0.3983 | 0.2590 | 0.3154 |
| | **CARD (Ours)** | **0.9580** | **0.9615** | **0.9375** | **0.9410** | **0.9280** | **0.9075** | **0.9144** | **0.8730** | **0.8721** | **0.8547** | **0.9101** | **0.9168** | **0.8837** | **0.8752** | **0.8488** |
| Scene-15 | PIC [IJCAI 2019] | 0.4513 | 0.3944 | 0.4475 | 0.4219 | 0.3913 | 0.4331 | 0.3980 | 0.4021 | 0.3760 | 0.3586 | 0.2796 | 0.2352 | 0.2649 | 0.2387 | 0.2183 |
| | EEIMVC [TPAMI 2021] | 0.4609 | 0.4388 | 0.3924 | 0.3648 | 0.4116 | 0.4250 | 0.3944 | 0.3488 | 0.3395 | 0.3340 | 0.2857 | 0.2681 | 0.2198 | 0.1997 | 0.2114 |
| | IMVC-CBG [CVPR 2022] | 0.3902 | 0.3833 | 0.3594 | 0.3494 | 0.3021 | 0.3532 | 0.3370 | 0.3012 | 0.2738 | 0.2520 | 0.2066 | 0.2167 | 0.1775 | 0.1802 | 0.1398 |
| | SIMVC-SA [MM 2023] | 0.3712 | 0.3365 | 0.3070 | 0.3068 | 0.2638 | 0.3199 | 0.2857 | 0.2571 | 0.2387 | 0.2207 | 0.1896 | 0.1627 | 0.1416 | 0.1236 | 0.1087 |
| | FCMVC-IVA [TIP 2024] | 0.2368 | 0.2488 | 0.2187 | 0.2201 | 0.2252 | 0.1683 | 0.1703 | 0.1521 | 0.1555 | 0.1357 | 0.0796 | 0.0835 | 0.0666 | 0.0727 | 0.0629 |
| | FIMVC-VIA [TNNLS 2024] | 0.4252 | 0.4109 | 0.3935 | 0.3338 | 0.3262 | 0.3904 | 0.3610 | 0.3369 | 0.2900 | 0.2698 | 0.2508 | 0.2292 | 0.2070 | 0.1618 | 0.1499 |
| | FSIMVC-OF [MM 2024] | 0.4319 | 0.4241 | 0.4036 | 0.3973 | 0.3788 | 0.4100 | 0.3856 | 0.3783 | 0.3507 | 0.3145 | 0.2518 | 0.2349 | 0.2328 | 0.2149 | 0.1854 |
| | LFIMVC [TIP 2024] | 0.3035 | 0.3052 | 0.2843 | 0.2631 | 0.2499 | 0.2419 | 0.2036 | 0.1851 | 0.1840 | 0.1644 | 0.1355 | 0.1144 | 0.0998 | 0.0979 | 0.0845 |
| | PSIMVC-PG [TNNLS 2024] | 0.3663 | 0.3527 | 0.3429 | 0.2678 | 0.2812 | 0.3338 | 0.2967 | 0.2909 | 0.2398 | 0.2326 | 0.1921 | 0.1714 | 0.1680 | 0.1243 | 0.1249 |
| | CAL [IJCAI 2025] | 0.1400 | 0.0912 | 0.0923 | 0.0914 | 0.0916 | 0.0762 | 0.0031 | 0.0036 | 0.0029 | 0.0030 | 0.0171 | 0.0002 | 0.0001 | 0.0001 | 0.0001 |
| | **CARD (Ours)** | **0.4725** | **0.4936** | **0.4762** | **0.4395** | **0.4528** | **0.4551** | **0.4590** | **0.4424** | **0.4063** | **0.3872** | **0.3065** | **0.3100** | **0.3033** | **0.2615** | **0.2656** |
| CCV | PIC [IJCAI 2019] | 0.1581 | 0.2450 | 0.2513 | 0.2174 | 0.1840 | 0.1215 | **0.2100** | 0.1850 | 0.1668 | 0.1329 | 0.0423 | 0.0997 | 0.0889 | 0.0811 | 0.0546 |
| | EEIMVC [TPAMI 2021] | 0.2469 | 0.2382 | 0.2204 | 0.2142 | 0.1818 | 0.1899 | 0.1786 | 0.1594 | 0.1515 | 0.1285 | 0.0887 | 0.0821 | 0.0711 | 0.0662 | 0.0519 |
| | IMVC-CBG [CVPR 2022] | 0.2169 | 0.2271 | 0.2107 | 0.1798 | 0.1682 | 0.1758 | 0.1706 | 0.1532 | 0.1324 | 0.1152 | 0.0706 | 0.0692 | 0.0553 | 0.0511 | 0.0421 |
| | SIMVC-SA [MM 2023] | 0.2410 | 0.2442 | 0.2197 | 0.1950 | 0.1884 | 0.1811 | 0.1721 | 0.1504 | 0.1378 | 0.1297 | 0.0826 | 0.0809 | 0.0656 | 0.0582 | 0.0534 |
| | FCMVC-IVA [TIP 2024] | 0.1627 | 0.1568 | 0.1671 | 0.1580 | 0.1652 | 0.1153 | 0.1106 | 0.1117 | 0.1047 | 0.1064 | 0.0385 | 0.0333 | 0.0266 | 0.0237 | 0.0209 |
| | FIMVC-VIA [TNNLS 2024] | 0.2531 | 0.2417 | 0.2311 | 0.2123 | 0.1897 | 0.1886 | 0.1732 | 0.1633 | 0.1513 | 0.1385 | 0.0870 | 0.0804 | 0.0727 | 0.0647 | 0.0552 |
| | FSIMVC-OF [MM 2024] | 0.2203 | 0.2175 | 0.2100 | 0.1853 | 0.1732 | 0.1637 | 0.1535 | 0.1389 | 0.1285 | 0.1202 | 0.0684 | 0.0647 | 0.0566 | 0.0507 | 0.0440 |
| | LFIMVC [TIP 2024] | 0.2035 | 0.2018 | 0.2064 | 0.1792 | 0.1789 | 0.1542 | 0.1336 | 0.1331 | 0.1159 | 0.1102 | 0.0643 | 0.0555 | 0.0564 | 0.0475 | 0.0435 |
| | PSIMVC-PG [TNNLS 2024] | 0.2027 | 0.1940 | 0.1882 | 0.1662 | 0.1587 | 0.1591 | 0.1455 | 0.1386 | 0.1193 | 0.1058 | 0.0637 | 0.0583 | 0.0566 | 0.0455 | 0.0400 |
| | CAL [IJCAI 2025] | 0.2263 | 0.1565 | 0.1398 | 0.1711 | 0.1639 | 0.1484 | 0.0901 | 0.0704 | 0.1065 | 0.0888 | 0.0901 | 0.0455 | 0.0233 | 0.0509 | 0.0463 |
| | **CARD (Ours)** | **0.2752** | **0.2688** | **0.2575** | **0.2345** | **0.2080** | **0.2097** | 0.1965 | **0.1884** | **0.1738** | **0.1572** | **0.1045** | **0.1003** | **0.0907** | **0.0851** | **0.0652** |
| SUNRGBD | PIC [IJCAI 2019] | 0.1653 | 0.1777 | 0.1863 | 0.1524 | 0.1496 | 0.2182 | **0.2356** | 0.2078 | 0.1946 | 0.1787 | 0.0802 | 0.0895 | 0.0881 | 0.0661 | 0.0626 |
| | EEIMVC [TPAMI 2021] | 0.1916 | 0.1792 | 0.1742 | 0.1532 | 0.1434 | 0.2168 | 0.2073 | 0.1862 | 0.1678 | 0.1578 | 0.0926 | 0.0819 | 0.0734 | 0.0578 | 0.0511 |
| | IMVC-CBG [CVPR 2022] | 0.2082 | 0.1901 | 0.1953 | 0.1819 | **0.1870** | 0.1600 | 0.1611 | 0.1635 | 0.0788 | 0.0921 | 0.0767 | 0.0833 | 0.0777 | 0.0392 | 0.0374 |
| | SIMVC-SA [MM 2023] | 0.1867 | 0.1752 | 0.1680 | 0.1501 | 0.1432 | 0.2237 | 0.2027 | 0.1897 | 0.1678 | 0.1561 | 0.0888 | 0.0799 | 0.0705 | 0.0579 | 0.0525 |
| | FCMVC-IVA [TIP 2024] | 0.1148 | 0.1115 | 0.1041 | 0.1010 | 0.0903 | 0.1366 | 0.1248 | 0.1089 | 0.1010 | 0.0950 | 0.0408 | 0.0372 | 0.0298 | 0.0283 | 0.0232 |
| | FIMVC-VIA [TNNLS 2024] | 0.1936 | 0.1852 | 0.1712 | 0.1625 | 0.1448 | 0.2254 | 0.2136 | 0.1940 | 0.1675 | 0.1567 | 0.0975 | 0.0876 | 0.0748 | 0.0609 | 0.0511 |
| | FSIMVC-OF [MM 2024] | 0.1874 | 0.1865 | 0.1728 | 0.1617 | 0.1539 | 0.1924 | 0.1838 | 0.1750 | 0.1639 | 0.1623 | 0.0813 | 0.0819 | 0.0666 | 0.0551 | 0.0444 |
| | LFIMVC [TIP 2024] | 0.1796 | 0.1785 | 0.1593 | 0.1429 | 0.1269 | 0.2065 | 0.2045 | 0.1770 | 0.1543 | 0.1325 | 0.0820 | 0.0753 | 0.0666 | 0.0511 | 0.0398 |
| | PSIMVC-PG [TNNLS 2024] | 0.1526 | 0.1566 | 0.1425 | 0.1310 | 0.1340 | 0.1865 | 0.1797 | 0.1610 | 0.1498 | 0.1349 | 0.0614 | 0.0637 | 0.0528 | 0.0475 | 0.0410 |
| | CAL [IJCAI 2025] | 0.2003 | 0.1684 | 0.1489 | 0.1416 | 0.1393 | 0.1604 | 0.1132 | 0.0709 | 0.0431 | 0.0443 | 0.0428 | 0.0310 | 0.0146 | 0.0092 | 0.0138 |
| | **CARD (Ours)** | **0.2171** | **0.1945** | **0.1962** | **0.1870** | **0.1624** | **0.2348** | 0.2117 | **0.2157** | **0.2002** | **0.1838** | **0.1118** | **0.0924** | **0.0938** | **0.0832** | **0.0683** |
| NUSWIDE-OBJ | PIC [IJCAI 2019] | – | – | – | – | – | – | – | – | – | – | – | – | – | – | – |
| | EEIMVC [TPAMI 2021] | 0.1427 | 0.1346 | 0.1369 | 0.1334 | 0.1264 | 0.1136 | 0.1161 | 0.1110 | 0.0979 | 0.0926 | 0.0488 | 0.0458 | 0.0466 | 0.0402 | 0.0369 |
| | IMVC-CBG [CVPR 2022] | 0.1498 | 0.1554 | **0.1585** | 0.1451 | 0.1427 | 0.1048 | 0.0894 | 0.0660 | 0.0653 | 0.0594 | 0.0299 | 0.0265 | 0.0207 | 0.0223 | 0.0255 |
| | SIMVC-SA [MM 2023] | 0.1213 | 0.1212 | 0.1190 | 0.1112 | 0.1101 | 0.0953 | 0.0901 | 0.0859 | 0.0794 | 0.0785 | 0.0333 | 0.0335 | 0.0307 | 0.0264 | 0.0265 |
| | FCMVC-IVA [TIP 2024] | – | – | – | – | – | – | – | – | – | – | – | – | – | – | – |
| | FIMVC-VIA [TNNLS 2024] | 0.1389 | 0.1335 | 0.1350 | 0.1253 | 0.1203 | 0.1121 | 0.1020 | 0.1012 | 0.0902 | 0.0871 | 0.0430 | 0.0391 | 0.0391 | 0.0345 | 0.0334 |
| | FSIMVC-OF [MM 2024] | 0.1394 | 0.1357 | 0.1420 | 0.1314 | 0.1242 | 0.1137 | 0.1124 | 0.1091 | 0.1029 | 0.0973 | 0.0481 | 0.0406 | 0.0417 | 0.0394 | 0.0353 |
| | LFIMVC [TIP 2024] | 0.1299 | 0.1194 | 0.1221 | 0.1130 | 0.1064 | 0.1005 | 0.0896 | 0.0854 | 0.0810 | 0.0724 | 0.0370 | 0.0299 | 0.0311 | 0.0278 | 0.0230 |
| | PSIMVC-PG [TNNLS 2024] | 0.1102 | 0.1133 | 0.1208 | 0.1124 | 0.1162 | 0.0964 | 0.0941 | 0.0919 | 0.0882 | 0.0838 | 0.0294 | 0.0316 | 0.0317 | 0.0280 | 0.0285 |
| | CAL [IJCAI 2025] | – | – | – | – | – | – | – | – | – | – | – | – | – | – | – |
| | **CARD (Ours)** | **0.1557** | **0.1604** | 0.1480 | **0.1490** | **0.1451** | **0.1520** | **0.1506** | **0.1379** | **0.1313** | **0.1253** | **0.0614** | **0.0682** | **0.0569** | **0.0536** | **0.0533** |

methods. The single-run runtime protocol is detailed in Appendix C.5.

## 5.5. Parameter Sensitivity Analysis

Figure 1 visualizes the ACC surfaces on NUSWIDEOBJ, which remain relatively flat across $\rho \in \{0.1, 0.5, 0.9\}$, indicating that CARD is not overly sensitive to hyperparameter variations across practical anchor budgets.

The performance surfaces remain relatively flat across a wide range of parameter choices, even under extreme missingness ($\rho = 0.9$), indicating that CARD is relatively stable

across the tested practical hyperparameter settings. Variation mainly appears under very small anchor budgets, where anchor coverage becomes sparse. Parameter sensitivity results on additional datasets are provided in the Appendix C.6.

## 5.6. Convergence Study

We track the objective value in Eq. (2) over outer iterations. Across all tested missing ratios, including the extreme setting with $\rho = 0.9$, the objective decreases rapidly in the first few outer iterations and then stabilizes. This empirical

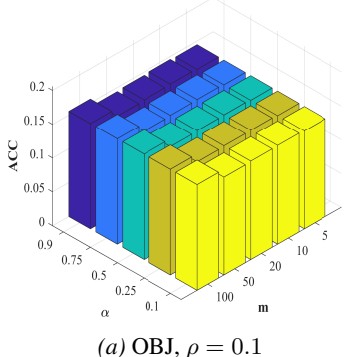
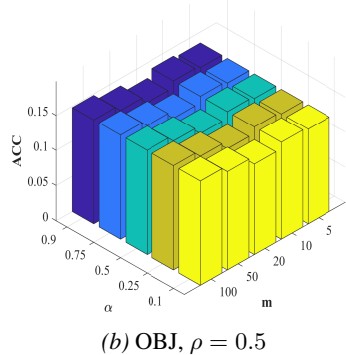
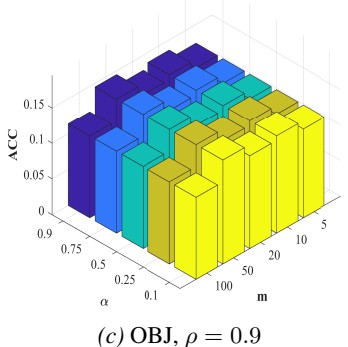

*(a)* OBJ, $\rho = 0.1$      *(b)* OBJ, $\rho = 0.5$      *(c)* OBJ, $\rho = 0.9$

*Figure 1.* Parameter sensitivity of CARD on NUSWIDEOBJ under different missing ratios $\rho \in \{0.1, 0.5, 0.9\}$, measured by ACC over the searched hyperparameter grid.

behavior supports stable self-consistency refinement.

We defer comprehensive convergence curves on all benchmark datasets and missing ratios to the Appendix C.7, where consistent and robust convergence behavior is observed across different data scales and incompleteness levels.

### 5.7. Ablation Study

We conduct ablation studies on *complete data* ($\rho = 0$) to isolate the contribution of the two key optimization components in CARD under a clean and interpretable setting: high-order resolvent diffusion and the $\mathbf{C} \leftrightarrow \mathbf{Y}$ feedback mechanism. Starting from an average anchor-kernel variant without high-order diffusion, we first add resolvent diffusion while keeping the anchor-kernel target fixed, and then enable the full $\mathbf{C} \leftrightarrow \mathbf{Y}$ feedback used by CARD. Detailed formulations of these variants are provided in the Appendix C.8.

Results are summarized in Table 2. The resolvent-diffusion variant improves the average anchor-kernel variant on most datasets and metrics, indicating the benefit of high-order structural propagation beyond simple kernel averaging. The full CARD model further couples diffusion-based affinity learning with clustering feedback and achieves the strongest overall performance, supporting the effectiveness of the joint design.

### 6. Conclusion

We proposed **CARD**, a framework that reframes incomplete multi-view clustering from a spectral filtering perspective rather than a data imputation paradigm. By operating on observed relational evidence, CARD treats missing views as the absence of admissible diffusion paths, without assigning artificial similarities or explicit completion. We introduced a unified anchor-induced hypergraph and high-order resolvent diffusion operator, which acts as a rational spectral filter to emphasize consensus structures relative to view-specific

*Table 2.* Ablation study on high-order resolvent diffusion and clustering feedback under complete data ($\rho = 0$).

| Dataset | Metric | Avg. Anchor Kernel | +Resolvent Diff. | **CARD** |
|---|---|---|---|---|
| UCIDigits | ACC | 0.8425 | 0.9330 | **0.9670** |
| | NMI | 0.8705 | 0.8994 | **0.9028** |
| | ARI | 0.7887 | 0.8601 | **0.9201** |
| Scene-15 | ACC | 0.4384 | 0.4629 | **0.4908** |
| | NMI | 0.4321 | 0.4420 | **0.4646** |
| | ARI | 0.2496 | 0.2782 | **0.3276** |
| CCV | ACC | 0.1733 | 0.1890 | **0.2984** |
| | NMI | 0.1344 | 0.1560 | **0.2173** |
| | ARI | 0.0402 | 0.0458 | **0.1117** |
| SUNRGBD | ACC | 0.2236 | **0.2394** | 0.2348 |
| | NMI | 0.2171 | 0.2094 | **0.2451** |
| | ARI | 0.0812 | 0.0753 | **0.1131** |
| NUSWIDE -OBJ | ACC | 0.1727 | 0.1723 | **0.1759** |
| | NMI | 0.1359 | 0.1360 | **0.1504** |
| | ARI | 0.0609 | 0.0606 | **0.0709** |

variations under partial observations. An implicit solver bridges high-order diffusion modeling with scalable optimization, reducing each fixed-kernel update to a Euclidean K-means step while avoiding dense $N \times N$ matrices. The theoretical analysis explains the spectral selectivity and stability properties of the resolvent update, while experiments show that CARD remains effective under the tested severe incompleteness settings and retains linear scaling in $N$ for a fixed anchor budget. These results suggest that principled operator design on observed relational structures offers an effective alternative to imputation-based strategies for incomplete multi-view learning.

### Acknowledgements

This work was supported by the National Natural Science Foundation of China (62376146, U2541227), and the Shanxi Provincial Central Guiding Fund for Local Science and Technology Development Special Project (YDZJSX20231D003).

## Impact Statement

This paper presents work whose goal is to advance the field of Machine Learning, specifically in robust data analysis under uncertainty. The proposed method, CARD, offers a resource-efficient solution for handling incomplete data without relying on computationally intensive generative models. This has potential positive impacts in scenarios with unreliable sensor networks, privacy-constrained data sharing, or low-resource computing environments where data quality cannot be guaranteed. We do not anticipate specific negative societal consequences beyond those generally associated with clustering methods, such as potential misuse of discovered groups.

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

## Preliminaries: Global Notation and Low-Rank Structure

**Purpose of this section.** This preliminary section is included solely to unify notation and spectral conventions used throughout the appendix. All quantities defined here are consistent with those introduced in Section 3 and Section 4 of the main text, and no additional modeling assumptions are imposed beyond those already stated.

In particular, this section formalizes the compact spectral representation implicitly used in the derivations of Appendix A and the theoretical analysis of Appendix B.

- **Anchor-Induced Operator.** We consider the compact SVD of the normalized anchor incidence matrix and the induced spectral factorization

$$\mathbf{Q} = \mathbf{U}_r \mathbf{\Sigma}_r \mathbf{V}_r^\top, \qquad \mathbf{A} = \mathbf{Q}\mathbf{Q}^\top = \mathbf{U}_r \mathbf{\Sigma}_r^2 \mathbf{U}_r^\top. \tag{24}$$

  Here, $r = \text{rank}(\mathbf{Q}) = \text{rank}(\mathbf{A})$, $\mathbf{U}_r \in \mathbb{R}^{N \times r}$ contains the $r$ nonzero eigenvectors of $\mathbf{A}$, and $\mathbf{\Sigma}_r^2 = \text{diag}(\sigma_1^2, \ldots, \sigma_r^2)$ collects the corresponding nonzero eigenvalues. When discussing eigenvalue responses of $\mathbf{A}$, we write $\lambda_i = \sigma_i^2$.

- **Spectral Normalization.** The operator $\mathbf{Q}$ is constructed from a degree-normalized bipartite anchor incidence graph, and $\mathbf{A} = \mathbf{Q}\mathbf{Q}^\top$ is symmetric and positive semi-definite. We use the normalized operator, or equivalently rescale it when necessary, so that $\|\mathbf{A}\|_2 \leq 1$. This is the normalization condition used in both the compact resolvent derivation and the theoretical analysis. Zero-degree anchors are removed before normalization, and each sample is assumed to have at least one observed view.

- **Spectral Separation.** The analysis allows $\mathbf{A}$ to have rank larger than $c$. Its leading $c$ eigenmodes are interpreted as cluster-level consensus modes, while the remaining $r - c$ modes capture residual variations induced by intra-cluster diversity and view-specific discrepancies. For interpretation, the nonzero eigenvalues of $\mathbf{A}$ may be ordered as

$$1 \geq \lambda_1 \geq \cdots \geq \lambda_c > \lambda_{c+1} \geq \cdots \geq \lambda_r > 0. \tag{25}$$

  The gap between $\lambda_c$ and $\lambda_{c+1}$ quantifies the degree of structural separation. We do not assume $\lambda_c = 1$; consensus modes are only assumed to be separated from residual modes when spectral-selectivity interpretations are invoked.

## A. Appendix A: Proofs for Methodology in Section 3

**Roadmap.** Appendix A establishes the mathematical foundations of the CARD framework in four steps: (i) we motivate a high-order relational consistency energy from anchor-induced evidence; (ii) we show how this energy leads to a diffusion-based regularization form; (iii) we characterize the resulting optimization via a resolvent operator in a compact spectral space; and (iv) we prove that the fixed-kernel assignment update is equivalent to a scalable K-means procedure on explicit features.

### A.1. Derivation of the High-Order Diffusion Regularization

A.1.1. HIGH-ORDER RELATIONAL CONSISTENCY ENERGY

We assume that the learned similarity matrix $\mathbf{C}$ is symmetric, i.e., $C_{ij} = C_{ji}$, which is standard in graph-based clustering formulations. Let $B_{ia}^{(v)}$ denote the affinity between sample $i$ and anchor $a$ in view $v$, and let $\Pi_{iv}$ indicate view availability. The availability-gated affinity is defined as $\hat{B}_{ia}^{(v)} = \Pi_{iv} B_{ia}^{(v)}$. This unnormalized energy is used only as a design motivation; the objective in the main text uses the normalized incidence $\tilde{\mathbf{H}}$ and the induced operator $\mathbf{A}$.

**Design rationale.** The following energy is constructed to encourage consistency between pairs of similarity entries $(C_{ij}, C_{i'j'})$ whenever the corresponding sample pairs $(i, i')$ and $(j, j')$ are supported by shared anchor-induced relations. This formulation generalizes first-order graph smoothness to a higher-order setting without relying on feature imputation:

$$\mathcal{E} = \frac{1}{2} \sum_{i,j=1}^{N} \sum_{i',j'=1}^{N} \sum_{v,v'=1}^{V} \sum_{a=1}^{m_v} \sum_{b=1}^{m_{v'}} \hat{B}_{ia}^{(v)} \hat{B}_{i'a}^{(v)} \hat{B}_{jb}^{(v')} \hat{B}_{j'b}^{(v')} \big( C_{ij} - C_{i'j'} \big)^2. \tag{26}$$

The factor $1/2$ plays the usual role of avoiding double counting in the symmetric pairwise smoothness expansion. Allowing $v \neq v'$ enables cross-view relational coupling, which is useful in incomplete multi-view settings.

### A.1.2. NORMALIZATION AND MATRIX FORM

Let $(v, a)$ index the concatenated anchor set across all views, and define the gated incidence $\tilde{H}_{i,(v,a)} = \hat{H}_{ia}^{(v)}$. The vertex and anchor degrees are

$$D_{ii} = \sum_{v,a} \tilde{H}_{i,(v,a)} W_{(v,a)(v,a)}, \tag{27}$$

$$(\mathbf{D}_e)_{(v,a)(v,a)} = \sum_i \tilde{H}_{i,(v,a)}. \tag{28}$$

The normalized bipartite operator is

$$\mathbf{Q} = \mathbf{D}^{-1/2}\tilde{\mathbf{H}}\mathbf{W}^{1/2}\mathbf{D}_e^{-1/2}, \qquad \mathbf{A} = \mathbf{Q}\mathbf{Q}^\top. \tag{29}$$

Under this construction, $\mathbf{A}$ is a symmetric normalized hypergraph diffusion kernel. In the incomplete-view case, missing views only remove corresponding terms without assigning artificial relations to unobserved views.

### A.1.3. REDUCTION TO DIFFUSION REGULARIZATION

Using the normalized diffusion weights induced by $\mathbf{A}$, we adopt the following normalized high-order Dirichlet surrogate:

$$\mathcal{E}_A(\mathbf{C}) = \operatorname{Tr}(\mathbf{C}^\top\mathbf{C}) - \operatorname{Tr}(\mathbf{C}^\top\mathbf{A}\mathbf{C}\mathbf{A}^\top). \tag{30}$$

When the diffusion weights are stochastic, this form coincides with the standard pairwise high-order smoothness expansion. In the general normalized case, it gives the corresponding degree-normalized counterpart used in Eq. (2). This characterization motivates interpreting the optimization as favoring similarity matrices that are stable under anchor-induced diffusion, leading to the resolvent formulation developed next.

## A.2. Derivation of the Resolvent Operator

**Role in the framework.** This subsection derives the relationship between the regularization parameter $\mu$ in Eq. (2) and the diffusion parameter $\alpha$ used in the resolvent formulation. This derivation supports the closed-form solution in Theorem 3.1 and the dynamic reformulation in Theorem 3.2.

**Lemma A.1** (Parameter Mapping)**.** *The regularization parameter $\mu$ in Eq. (2) and the diffusion parameter $\alpha$ in the resolvent operator satisfy $\alpha = 1/(1 + \mu)$.*

*Proof.* The fixed-$\mathbf{Y}$ stationary condition is

$$(1 + \mu)\mathbf{C} - \mathbf{A}\mathbf{C}\mathbf{A}^\top = \mu\mathbf{S}. \tag{31}$$

Dividing both sides by $1 + \mu$ gives

$$\mathbf{C} - \frac{1}{1 + \mu}\mathbf{A}\mathbf{C}\mathbf{A}^\top = \frac{\mu}{1 + \mu}\mathbf{S}. \tag{32}$$

By defining $\alpha \triangleq 1/(1 + \mu)$, we have $1 - \alpha = \mu/(1 + \mu)$, and thus

$$\mathbf{C} - \alpha\mathbf{A}\mathbf{C}\mathbf{A}^\top = (1 - \alpha)\mathbf{S}. \tag{33}$$

Applying $\operatorname{vec}(\cdot)$ and using $\operatorname{vec}(\mathbf{A}\mathbf{C}\mathbf{B}) = (\mathbf{B}^\top \otimes \mathbf{A})\operatorname{vec}(\mathbf{C})$, we obtain

$$(\mathbf{I} - \alpha\mathbf{A} \otimes \mathbf{A})\operatorname{vec}(\mathbf{C}) = (1 - \alpha)\operatorname{vec}(\mathbf{S}). \tag{34}$$

Since $\alpha \in (0, 1)$ and $\|\mathbf{A}\|_2 \leq 1$, the inverse exists, yielding

$$\operatorname{vec}(\mathbf{C}^*) = (1 - \alpha)(\mathbf{I} - \alpha\mathbf{A} \otimes \mathbf{A})^{-1}\operatorname{vec}(\mathbf{S}). \tag{35}$$

$\square$

*Remark* A.2. As $\mu \to 0$, $\alpha \to 1$, leading to increasingly long-range diffusion. In practice, $\alpha$ is chosen strictly below 1 to balance diffusion strength and numerical conditioning, consistent with Appendix B.4.

## A.3. Proof of Theorem 3.1: Compact Spectral Filtering

**Motivation.** Solving the Sylvester equation on the full $N \times N$ matrix is computationally prohibitive. We show that the low-rank structure of $\mathbf{A}$ allows the solution $\mathbf{C}^*$ to be represented in the compact anchor-induced spectral subspace.

*Proof.* Starting from Eq. (33), substitute the compact factorization $\mathbf{A} = \mathbf{U}_r \mathbf{\Sigma}_r^2 \mathbf{U}_r^\top$. The operator $\mathbf{A}$ acts non-trivially only on $\mathrm{span}(\mathbf{U}_r)$, while all components not affected on both sides by $\mathrm{span}(\mathbf{U}_r)$ pass through the equation with coefficient $1 - \alpha$.

Projecting the equation onto $\mathrm{span}(\mathbf{U}_r)$ gives

$$\mathbf{U}_r^\top \mathbf{C} \mathbf{U}_r - \alpha \mathbf{\Sigma}_r^2 (\mathbf{U}_r^\top \mathbf{C} \mathbf{U}_r) \mathbf{\Sigma}_r^2 = (1 - \alpha) \mathbf{U}_r^\top \mathbf{S} \mathbf{U}_r. \tag{36}$$

Let $\mathbf{X} = \mathbf{U}_r^\top \mathbf{C} \mathbf{U}_r$ and $\mathbf{M} = \mathbf{U}_r^\top \mathbf{S} \mathbf{U}_r$. Then, element-wise,

$$X_{ij} - \alpha \sigma_i^2 \sigma_j^2 X_{ij} = (1 - \alpha) M_{ij}, \tag{37}$$

so

$$X_{ij} = \frac{1 - \alpha}{1 - \alpha \sigma_i^2 \sigma_j^2} M_{ij}. \tag{38}$$

Define $\mathbf{\Omega}_r \in \mathbb{R}^{r \times r}$ by

$$[\mathbf{\Omega}_r]_{ij} = \frac{\alpha(1 - \alpha)\sigma_i^2 \sigma_j^2}{1 - \alpha \sigma_i^2 \sigma_j^2}. \tag{39}$$

Combining the anchor-subspace correction with the residual pass-through term yields

$$\mathbf{C}^* = (1 - \alpha)\mathbf{S} + \mathbf{U}_r \left( \mathbf{\Omega}_r \circ (\mathbf{U}_r^\top \mathbf{S} \mathbf{U}_r) \right) \mathbf{U}_r^\top. \tag{40}$$

$\square$

*Remark* A.3. This result reduces the effective spectral computation from the full $N$-dimensional space to the $r$-dimensional anchor-induced subspace.

## A.4. Proof of PSD Condition and Factorization Existence

**Motivation.** Algorithm 1 employs a factorization of the reduced spectral filter $\mathbf{\Omega}_r$ to construct explicit features. We show that $\mathbf{\Omega}_r$ is positive semi-definite.

*Proof.* Recall that

$$[\mathbf{\Omega}_r]_{ij} = \frac{\alpha(1 - \alpha)\sigma_i^2 \sigma_j^2}{1 - \alpha \sigma_i^2 \sigma_j^2}, \qquad 1 \le i, j \le r, \tag{41}$$

where $\alpha \in (0, 1)$ and $\sigma_i^2 \in [0, 1]$. Since $0 \le \alpha \sigma_i^2 \sigma_j^2 < 1$, the denominator admits the absolutely convergent expansion

$$\frac{1}{1 - \alpha \sigma_i^2 \sigma_j^2} = \sum_{k=0}^{\infty} (\alpha \sigma_i^2 \sigma_j^2)^k. \tag{42}$$

Thus,

$$[\mathbf{\Omega}_r]_{ij} = \sum_{k=0}^{\infty} \alpha(1 - \alpha)\alpha^k \sigma_i^{2(k+1)} \sigma_j^{2(k+1)}. \tag{43}$$

Let $a_i = \sigma_i^2$ and define $\mathbf{v}^{(k)} \in \mathbb{R}^r$ by

$$v_i^{(k)} = \sqrt{\alpha(1 - \alpha)}\, \alpha^{k/2} a_i^{k+1}. \tag{44}$$

Then

$$\mathbf{\Omega}_r = \sum_{k=0}^{\infty} \mathbf{v}^{(k)} (\mathbf{v}^{(k)})^\top. \tag{45}$$

Each summand is positive semi-definite, and the series converges absolutely entry-wise. Therefore, $\mathbf{\Omega}_r \succeq 0$. $\square$

*Remark* A.4. The PSD property guarantees the existence of a factorization $\mathbf{\Omega}_r = \mathbf{G}_r \mathbf{G}_r^\top$. The algorithm does not require computing the infinite series and is independent of the particular PSD factorization used.

## A.5. Proof of Theorem 3.2: Iterative Exact Reformulation

**Motivation.** We show that the fixed-kernel trace maximization on the dense similarity matrix $\mathbf{C}^{(t)}$ is exactly equivalent to Euclidean K-means on an explicit feature embedding.

*Proof.* At iteration $t$, consider the kernel K-means objective

$$\mathcal{J}^{(t)}(\tilde{\mathbf{Y}}) = \mathrm{Tr}\left(\tilde{\mathbf{Y}}^\top \mathbf{C}^{(t)} \tilde{\mathbf{Y}}\right), \tag{46}$$

where $\mathbf{C}^{(t)}$ is constructed from the previous assignment $\tilde{\mathbf{Y}}^{(t)}$ and is fixed when optimizing with respect to $\tilde{\mathbf{Y}}$.

From Theorem 3.1,

$$\mathbf{C}^{(t)} = (1 - \alpha)\mathbf{S}^{(t)} + \mathbf{U}_r \left(\mathbf{\Omega}_r \circ (\mathbf{U}_r^\top \mathbf{S}^{(t)} \mathbf{U}_r)\right) \mathbf{U}_r^\top, \tag{47}$$

where $\mathbf{S}^{(t)} = \tilde{\mathbf{Y}}^{(t)} \tilde{\mathbf{Y}}^{(t)\top}$.

The first term in Eq. (47) is a Gram matrix:

$$(1 - \alpha)\mathbf{S}^{(t)} = \left(\sqrt{1 - \alpha}\,\tilde{\mathbf{Y}}^{(t)}\right) \left(\sqrt{1 - \alpha}\,\tilde{\mathbf{Y}}^{(t)}\right)^\top. \tag{48}$$

Define $\mathbf{J}^{(t)} = \mathbf{U}_r^\top \tilde{\mathbf{Y}}^{(t)}$. Since $\mathbf{\Omega}_r = \mathbf{G}_r \mathbf{G}_r^\top$ and $\mathbf{G}_r = [\mathbf{g}_1, \ldots, \mathbf{g}_r]$, we have

$$(\mathbf{g}_i \mathbf{g}_i^\top) \circ (\mathbf{J}^{(t)} \mathbf{J}^{(t)\top}) = (\mathrm{diag}(\mathbf{g}_i)\mathbf{J}^{(t)})(\mathrm{diag}(\mathbf{g}_i)\mathbf{J}^{(t)})^\top. \tag{49}$$

Summing over $i$ gives

$$\mathbf{\Omega}_r \circ (\mathbf{J}^{(t)} \mathbf{J}^{(t)\top}) = \sum_{i=1}^{r} (\mathrm{diag}(\mathbf{g}_i)\mathbf{J}^{(t)})(\mathrm{diag}(\mathbf{g}_i)\mathbf{J}^{(t)})^\top. \tag{50}$$

Define

$$\mathbf{P}^{(t)} = \left[\mathrm{diag}(\mathbf{g}_1)\mathbf{J}^{(t)}, \ldots, \mathrm{diag}(\mathbf{g}_r)\mathbf{J}^{(t)}\right] \in \mathbb{R}^{r \times rc}. \tag{51}$$

Then the filtered term is

$$\mathbf{C}_{\mathrm{filter}}^{(t)} = (\mathbf{U}_r \mathbf{P}^{(t)})(\mathbf{U}_r \mathbf{P}^{(t)})^\top. \tag{52}$$

Combining the two Gram factors, we obtain

$$\mathbf{C}^{(t)} = \mathbf{Z}^{(t)} \mathbf{Z}^{(t)\top}, \qquad \mathbf{Z}^{(t)} = \left[\mathbf{U}_r \mathbf{P}^{(t)}, \sqrt{1 - \alpha}\,\tilde{\mathbf{Y}}^{(t)}\right]. \tag{53}$$

Therefore, with $\mathbf{C}^{(t)}$ fixed, maximizing $\mathcal{J}^{(t)}(\tilde{\mathbf{Y}})$ is equivalent to minimizing the Euclidean K-means objective on $\mathbf{Z}^{(t)}$, up to a fixed constant. This proves the fixed-kernel reformulation. $\square$

## A.6. Algorithm Details

We provide the complete pseudocode for CARD in Algorithm 1. The algorithm consists of a one-shot initialization phase and a resolvent self-consistency refinement loop. In implementation, the $\mathbf{Y}$-update is reduced to Euclidean K-means and carried out by warm-started Lloyd iterations initialized from the previous partition. Empty clusters, if any, are reinitialized following standard K-means practice.

The use of $\mathbf{V}_r$ and $\mathbf{\Phi}$ follows from the compact SVD $\mathbf{Q} = \mathbf{U}_r \mathbf{\Sigma}_r \mathbf{V}_r^\top$. Since $\mathbf{\Sigma}_r^{-1}$ is applied only to the retained nonzero singular values, we have $\mathbf{Q}\mathbf{\Phi} = \mathbf{U}_r$. This allows the feature construction to be implemented without explicitly forming $\mathbf{U}_r$.

# B. Appendix B: Proofs for Theoretical Analysis in Section 4

This section proves the theoretical statements in Section 4. The results are properties of the fixed-$\mathbf{S}$ resolvent map, its spectral response, and its local stability. They do not constitute a global convergence or global recovery theorem for the full alternating discrete optimization procedure.

---

**Algorithm 1** CARD: Scalable Incomplete Multi-View Clustering

---

1: **Input:** View-specific gated bipartite graphs $\{\hat{\mathbf{H}}^{(v)}\}_{v=1}^{V}$, cluster number $c$, diffusion parameter $\alpha$
2: **Phase I: One-shot Initialization**
3: Construct $\mathbf{Q} \in \mathbb{R}^{N \times M}$ via Eq. (1)
4: Compute the compact SVD through $\mathbf{Q}^{\top}\mathbf{Q} = \mathbf{V}_r \mathbf{\Sigma}_r^2 \mathbf{V}_r^{\top}$
5: Compute $\mathbf{\Omega}_r$ and factorize $\mathbf{\Omega}_r = \mathbf{G}_r \mathbf{G}_r^{\top}$
6: Precompute $\mathbf{\Phi} = \mathbf{V}_r \mathbf{\Sigma}_r^{-1}$, where $\mathbf{\Sigma}_r^{-1}$ is taken over the retained nonzero singular values
7: Initialize $\mathbf{Y}^{(0)}$
8: **Phase II: Self-Consistency Refinement**
9: **repeat**
10:     Compute $\mathbf{J} \leftarrow \mathbf{\Phi}^{\top}(\mathbf{Q}^{\top}\tilde{\mathbf{Y}}^{(t)})$
11:     Compute $\mathbf{P}(\mathbf{Y}) \leftarrow [\text{diag}(\mathbf{g}_1)\mathbf{J}, \dots, \text{diag}(\mathbf{g}_r)\mathbf{J}]$
12:     Project features: $\mathbf{T} \leftarrow \mathbf{\Phi}\mathbf{P}(\mathbf{Y})$
13:     Construct $\mathbf{Z}^{(t)} \leftarrow [\mathbf{Q}\mathbf{T}, \sqrt{1-\alpha}\,\tilde{\mathbf{Y}}^{(t)}]$
14:     Update $\mathbf{Y}^{(t+1)}$ by K-means on $\mathbf{Z}^{(t)}$
15: **until** convergence
16: **Output:** Final cluster assignment $\mathbf{Y}$

---

## B.1. Proof of Proposition 4.2: Spectral Selectivity

**Motivation.** We quantify how the resolvent operator changes the relative gain of consensus and residual modes.

*Proof.* The spectral response of the resolvent operator on a diagonal Kronecker mode induced by eigenvalue $\lambda$ of $\mathbf{A}$ is

$$g_\alpha(\lambda) = \frac{1-\alpha}{1-\alpha\lambda^2}. \tag{54}$$

For $1 \geq \lambda_{\text{sig}} > \lambda_{\text{var}} \geq 0$, the selectivity ratio is

$$\text{SSR}(\mathcal{R}_\alpha) = \frac{g_\alpha(\lambda_{\text{sig}})}{g_\alpha(\lambda_{\text{var}})} = \frac{1-\alpha\lambda_{\text{var}}^2}{1-\alpha\lambda_{\text{sig}}^2}. \tag{55}$$

Since $\lambda_{\text{sig}} > \lambda_{\text{var}}$, this ratio is greater than one. Its derivative with respect to $\alpha$ is

$$\frac{d}{d\alpha}\text{SSR}(\mathcal{R}_\alpha) = \frac{\lambda_{\text{sig}}^2 - \lambda_{\text{var}}^2}{(1-\alpha\lambda_{\text{sig}}^2)^2} \geq 0. \tag{56}$$

If $\lambda_{\text{sig}} = 1$, then $\text{SSR}(\mathcal{R}_\alpha) = \Theta((1-\alpha)^{-1})$. Otherwise, taking $\alpha \to 1^-$ gives the finite limit $(1-\lambda_{\text{var}}^2)/(1-\lambda_{\text{sig}}^2)$. $\quad\square$

## B.2. Proof of Proposition 4.3: Path-Supported Diffusion Contribution

**Motivation.** We prove that anchor-supported paths provide a positive contribution to the fixed-$\mathbf{S}$ resolvent similarity.

*Proof.* The resolvent solution admits the Neumann expansion

$$\mathbf{C} = (1-\alpha)\sum_{k=0}^{\infty} \alpha^k \mathbf{A}^k \mathbf{S}(\mathbf{A}^k)^{\top}. \tag{57}$$

Because $\mathbf{A}$ and $\mathbf{S}$ are entrywise nonnegative, every term in the series is entrywise nonnegative. Isolating the $k = \ell$ term gives

$$\begin{aligned}
C_{ij} &\geq (1-\alpha)\alpha^\ell \left[\mathbf{A}^\ell \mathbf{S}(\mathbf{A}^\ell)^{\top}\right]_{ij} \\
&= (1-\alpha)\alpha^\ell \sum_{p',q'} (\mathbf{A}^\ell)_{ip'} S_{p'q'} (\mathbf{A}^\ell)_{jq'}.
\end{aligned} \tag{58}$$

Using the specific indices $p, q$ satisfying $(\mathbf{A}^\ell)_{ip} \geq \eta_i$, $(\mathbf{A}^\ell)_{jq} \geq \eta_j$, and $S_{pq} \geq s_{\min}$, we obtain

$$C_{ij} \geq (1 - \alpha)\alpha^\ell \eta_i \eta_j s_{\min} > 0. \tag{59}$$

This proves the claim. □

### B.3. Proof of Proposition 4.4: Conditional Local Subspace Refinement

**Motivation.** This proof records how the stated local assumptions imply the refinement recursion. The result is conditional and should not be read as a global convergence theorem.

*Proof.* Let $d_t = d(\mathcal{V}^{(t)}, \mathcal{V}^*) = \|\sin\Theta(\mathcal{V}^{(t)}, \mathcal{V}^*)\|_F$. By assumption, the continuous resolvent update satisfies

$$d_{\text{cont}}^{(t+1)} \leq C_{\text{loc}}\nu(\alpha)d_t. \tag{60}$$

The subsequent K-means discretization maps the continuous subspace to a discrete partition-induced subspace. By the bounded discretization assumption, this step introduces at most $\xi_{\text{disc}}$ additional subspace error. Therefore,

$$d(\mathcal{V}^{(t+1)}, \mathcal{V}^*) \leq d_{\text{cont}}^{(t+1)} + \xi_{\text{disc}} \leq C_{\text{loc}}\nu(\alpha)d(\mathcal{V}^{(t)}, \mathcal{V}^*) + \xi_{\text{disc}}. \tag{61}$$

If $C_{\text{loc}}\nu(\alpha) < 1$, this recursion is locally contractive up to the discretization error. □

*Remark* B.1. The factor $\nu(\alpha)$ comes from the gain ratio between residual and signal modes. The constant $C_{\text{loc}}$ absorbs local perturbation geometry, including eigengap and basin-dependent constants. Thus, the result is a conditional refinement statement rather than a direct consequence of Davis–Kahan alone.

### B.4. Proof of Proposition 4.5: Optimization Stability

**Motivation.** We bound the conditioning of the implicit linear system associated with the resolvent update.

*Proof.* The implicit solution involves the linear operator $\mathcal{L} = \mathbf{I} - \alpha(\mathbf{A} \otimes \mathbf{A})$. Since $\mathbf{A} = \mathbf{Q}\mathbf{Q}^\top \succeq 0$, the eigenvalues of $\mathbf{A} \otimes \mathbf{A}$ are $\lambda_i \lambda_j$, where $0 \leq \lambda_i, \lambda_j \leq \lambda_1 = \|\mathbf{A}\|_2 \leq 1$. Hence the eigenvalues of $\mathcal{L}$ lie in

$$[\, 1 - \alpha\lambda_1^2,\ 1\,]. \tag{62}$$

Therefore,

$$\kappa_2(\mathcal{L}) \leq \frac{1}{1 - \alpha\lambda_1^2} \leq \frac{1}{1 - \alpha}. \tag{63}$$

If $\mathbf{A}$ has a zero eigenvalue, then 1 is an eigenvalue of $\mathcal{L}$, and the first inequality becomes equality. □

*Remark* B.2. The bound shows that increasing $\alpha$ may worsen conditioning. If $\lambda_1 = 1$, the upper bound diverges as $\alpha \to 1^-$; if $\lambda_1 < 1$, the condition number remains finite but still increases with $\alpha$.

### B.5. Proof of Proposition 4.6: Stability to Anchor Perturbations

**Motivation.** This proof quantifies how perturbations in the anchor embedding $\mathbf{Q}$ affect the fixed-$\mathbf{S}$ resolvent similarity matrix. It does not consider the discontinuity of the subsequent discrete assignment step.

*Proof.* Recall that the fixed-$\mathbf{S}$ resolvent solution is

$$\text{vec}(\mathbf{C}^*) = (1 - \alpha)(\mathbf{I} - \alpha\mathbf{A} \otimes \mathbf{A})^{-1}\text{vec}(\mathbf{S}), \qquad \mathbf{A} = \mathbf{Q}\mathbf{Q}^\top. \tag{64}$$

Let the perturbed anchor embedding be $\hat{\mathbf{Q}} = \mathbf{Q} + \Delta\mathbf{Q}$, with $\|\Delta\mathbf{Q}\|_2 \leq \epsilon$. Then

$$\hat{\mathbf{A}} = \hat{\mathbf{Q}}\hat{\mathbf{Q}}^\top = \mathbf{A} + (\Delta\mathbf{Q})\mathbf{Q}^\top + \mathbf{Q}(\Delta\mathbf{Q})^\top + (\Delta\mathbf{Q})(\Delta\mathbf{Q})^\top. \tag{65}$$

Thus, to first order,

$$\Delta\mathbf{A} \triangleq \hat{\mathbf{A}} - \mathbf{A} = (\Delta\mathbf{Q})\mathbf{Q}^\top + \mathbf{Q}(\Delta\mathbf{Q})^\top + \mathcal{O}(\epsilon^2), \tag{66}$$

and

$$\|\Delta \mathbf{A}\|_2 \leq 2\|\mathbf{Q}\|_2 \epsilon + \mathcal{O}(\epsilon^2). \tag{67}$$

Define

$$\mathcal{R}_\alpha = (1 - \alpha)(\mathbf{I} - \alpha \mathbf{A} \otimes \mathbf{A})^{-1}, \tag{68}$$

$$\hat{\mathcal{R}}_\alpha = (1 - \alpha)(\mathbf{I} - \alpha \hat{\mathbf{A}} \otimes \hat{\mathbf{A}})^{-1}. \tag{69}$$

Using the first-order expansion of the resolvent identity,

$$\hat{\mathcal{R}}_\alpha - \mathcal{R}_\alpha = (1 - \alpha)(\mathbf{I} - \alpha \mathbf{A} \otimes \mathbf{A})^{-1} \alpha (\Delta \mathbf{A} \otimes \mathbf{A} + \mathbf{A} \otimes \Delta \mathbf{A})$$
$$\times (\mathbf{I} - \alpha \mathbf{A} \otimes \mathbf{A})^{-1} + \mathcal{O}(\epsilon^2). \tag{70}$$

Because $\|(\mathbf{I} - \alpha \mathbf{A} \otimes \mathbf{A})^{-1}\|_2 \leq (1 - \alpha)^{-1}$, we obtain

$$\|\hat{\mathcal{R}}_\alpha - \mathcal{R}_\alpha\|_2 \leq \frac{\alpha}{1 - \alpha}\|\Delta \mathbf{A} \otimes \mathbf{A} + \mathbf{A} \otimes \Delta \mathbf{A}\|_2 + \mathcal{O}(\epsilon^2). \tag{71}$$

Using $\|\mathbf{A}\|_2 = \|\mathbf{Q}\|_2^2$ and Eq. (67),

$$\|\Delta \mathbf{A} \otimes \mathbf{A} + \mathbf{A} \otimes \Delta \mathbf{A}\|_2 \leq 2\|\Delta \mathbf{A}\|_2 \|\mathbf{A}\|_2$$
$$\leq 4\|\mathbf{Q}\|_2^3 \epsilon + \mathcal{O}(\epsilon^2). \tag{72}$$

Finally,

$$\|\hat{\mathbf{C}} - \mathbf{C}^*\|_F \leq \|\hat{\mathcal{R}}_\alpha - \mathcal{R}_\alpha\|_2 \|\mathbf{S}\|_F$$
$$\leq \frac{4\alpha\|\mathbf{Q}\|_2^3}{1 - \alpha}\epsilon\|\mathbf{S}\|_F + \mathcal{O}(\epsilon^2). \tag{73}$$

This proves the claimed first-order bound. $\square$

## C. Appendix C: Experiments

### C.1. Datasets

We evaluate CARD on eight public multi-view benchmarks (Table 3), covering images, videos, and RGB-D scenes. In terms of applications, 100Leaves is used for leaf recognition (Cope et al., 2013); CCV for consumer video classification (Jiang et al., 2011); UCI Digits for handwritten digit recognition (Duin, 1998); NUSWIDE for web image retrieval (Chua et al., 2009); NUSWIDEOBJ for large-scale object-centric web image understanding and retrieval (Zhang et al., 2019); Scene-15 for scene category recognition (Lazebnik et al., 2006); SUNRGBD for RGB-D scene understanding (Song et al., 2015); Yale (Extended Yale B) for face recognition (Georghiades et al., 2001).

*Table 3.* Datasets used in the experiments. $V$ is the number of views, $n$ is the number of samples, and $c$ is the number of clusters.

| Dataset | $V$ | $n$ | $c$ |
|---|---|---|---|
| Yale | 3 | 165 | 15 |
| 100Leaves | 3 | 1,600 | 100 |
| NUSWIDE | 5 | 2,000 | 31 |
| UCI Digits | 6 | 2,000 | 10 |
| Scene-15 | 3 | 4,485 | 15 |
| CCV | 3 | 6,773 | 20 |
| SUNRGBD | 2 | 10,335 | 45 |
| NUSWIDEOBJ | 5 | 30,000 | 31 |

## C.2. Baselines

**Incomplete baselines.** We compare CARD with ten representative incomplete multi-view clustering (IMVC) methods: PIC (Wang et al., 2019), EEIMVC (Liu et al., 2021), IMVC-CBG (Wang et al., 2022c), SIMVC-SA (Wen et al., 2023b), FCMVC-IVA (Wan et al., 2024), FSIMVC-OF (Du et al., 2024), LFIMVC (Liu et al., 2019), PSIMVC-PG (Li et al., 2024b), CAL (Liu et al., 2025), and FIMVC-VIA (Liu et al., 2024b). These baselines cover spectral-perturbation modeling for incompleteness, regularized incomplete multi-view clustering, large-scale consensus bipartite graph learning, scalable anchor-based clustering with structure alignment, continual IMVC with incomplete views, duality optimal graph filtering for fast IMVC, late-fusion incomplete MVC, parameter-free prototype-graph based scalable IMVC, consensus-guided cross-view affinities learning, and fast IMVC with view-independent anchors, respectively.

**Complete baselines.** We also compare CARD with ten representative multi-view clustering methods under complete settings: EOMSC-CA (Liu et al., 2022), FSSC (Wang et al., 2022a), FPMVS-CAG (Wang et al., 2022d), FCAG (Nie et al., 2024), UDBGL (Fang et al., 2024), EDCAG (Wang et al., 2024c), BGAE (Li et al., 2024a), SSA-MVC (Wang et al., 2025a), ALPC (Chen et al., 2025), and AEVC (Liu et al., 2024a). These baselines cover one-pass multi-view subspace clustering with consensus anchors, fast self-supervised clustering with anchor graphs, parameter-free multi-view subspace clustering via consensus anchor guidance, fast clustering with anchor guidance, unified and discrete bipartite graph learning, efficient discrete clustering with anchor graphs, auto-encoding multi-view bipartite graph clustering, simple one-step multi-view clustering with fast similarity and cluster-structure learning, constrained anchor-based clustering, and doubly stochastic graph fusion, respectively.

## C.3. Additional Results under Incomplete Settings

This section provides additional experimental results and analysis under incomplete settings. Table 4 reports clustering performance on additional small- and medium-scale datasets, while Table 1 contains the datasets reported in the main text.

Across small- and medium-scale datasets, CARD achieves the best or competitive results in most reported settings over a wide range of missing rates. The results in Table 4 also show stable performance even when the missing rate reaches $\rho = 0.9$.

These empirical trends are consistent with the design of CARD. Availability gating reduces local relational support but does not introduce spurious cross-view affinities. By constructing a unified anchor hypergraph and performing high-order diffusion, CARD can use multi-hop anchor-supported relational evidence under severe missingness. Moreover, diffusion regularization further stabilizes the learned similarities, contributing to robust performance across a wide range of missing settings.

## C.4. Detailed Results under Complete Settings

Table 5 presents a comparison under the complete setting ($\rho = 0$) across eight multi-view datasets. CARD achieves the best or competitive performance on many dataset-metric pairs, indicating that the proposed structural diffusion design remains effective when all views are fully observed. CARD obtains top results on several datasets, including Yale, 100Leaves, UCIDigits, Scene-15, and CCV. On more challenging datasets such as NUSWIDE, SUNRGBD, and NUSWIDEOBJ, CARD remains competitive, although a few dataset-metric pairs favor other methods. These results suggest that the gains observed under incomplete settings are not achieved at the expense of complete-view performance.

## C.5. Runtime Analysis

Figure 2 reports the single-run execution time comparison on the SUNRGBD dataset, excluding hyperparameter search. All methods are evaluated under the same runtime environment. CARD completes the clustering task in $70s$, showing competitive efficiency among anchor-based methods and being much faster than dense graph-based methods such as PIC and CAL, which require more than $10^4$ seconds under the same setting. The strong baseline SSA-MVC is omitted because it triggered an out-of-memory error in our environment.

## C.6. Additional Parameter Analysis

This subsection provides a comprehensive parameter sensitivity analysis for datasets that are not included in the main text due to page constraints. We report ACC performance surfaces over the searched hyperparameter grid for all benchmark

*Table 4.* Clustering performance on small- to medium-scale datasets under incomplete settings (missing rates 0.1–0.9).

| Dataset | Method | ACC | | | | | NMI | | | | | ARI | | | | |
|---|---|---|---|---|---|---|---|---|---|---|---|---|---|---|---|---|
| | | 0.1 | 0.3 | 0.5 | 0.7 | 0.9 | 0.1 | 0.3 | 0.5 | 0.7 | 0.9 | 0.1 | 0.3 | 0.5 | 0.7 | 0.9 |
| Yale | PIC [IJCAI 2019] | 0.6303 | 0.5939 | 0.6727 | 0.6364 | 0.6242 | 0.6745 | 0.6587 | 0.6749 | 0.6652 | 0.6671 | 0.4577 | 0.4380 | 0.4612 | 0.4496 | 0.4547 |
| | EEIMVC [TPAMI 2021] | 0.7152 | 0.7091 | 0.6909 | 0.7091 | 0.6545 | 0.7210 | 0.7143 | 0.7086 | 0.7162 | 0.6570 | 0.5381 | 0.5356 | 0.5124 | 0.5183 | 0.4543 |
| | IMVC-CBG [CVPR 2022] | 0.6606 | 0.6242 | 0.6970 | 0.6545 | 0.6667 | 0.6491 | 0.6576 | 0.6909 | 0.6720 | 0.6582 | 0.4063 | 0.4251 | 0.4787 | 0.4527 | 0.4531 |
| | SIMVC-SA [MM 2023] | 0.7333 | 0.6667 | 0.6970 | 0.6667 | 0.6364 | 0.6978 | 0.6893 | 0.7028 | 0.6818 | 0.6321 | 0.5051 | 0.4967 | 0.5121 | 0.4765 | 0.4137 |
| | FCMVC-IVA [TIP 2024] | 0.6909 | 0.5636 | 0.5697 | 0.6121 | 0.4848 | 0.7020 | 0.5824 | 0.6167 | 0.6466 | 0.5011 | 0.5065 | 0.3504 | 0.3586 | 0.4234 | 0.2353 |
| | FIMVC-VIA [TNNLS 2024] | 0.7273 | 0.7243 | **0.7152** | 0.7152 | 0.6667 | 0.7235 | **0.7296** | **0.7151** | 0.7188 | 0.6576 | 0.5415 | **0.5440** | **0.5442** | 0.5379 | 0.4573 |
| | FSIMVC-OF [MM 2024] | 0.6242 | 0.6424 | 0.5636 | 0.6545 | 0.5879 | 0.6670 | 0.6697 | 0.6199 | 0.6955 | 0.6213 | 0.4696 | 0.4417 | 0.3978 | 0.4943 | 0.3902 |
| | LFIMVC [TIP 2024] | 0.6667 | 0.7212 | 0.6061 | 0.6727 | 0.5697 | 0.6692 | 0.7044 | 0.6146 | 0.6638 | 0.5794 | 0.4579 | 0.4969 | 0.3767 | 0.4523 | 0.3370 |
| | PSIMVC-PG [TNNLS 2024] | 0.7030 | 0.5939 | 0.6667 | 0.6667 | 0.6727 | 0.7093 | 0.6178 | 0.6724 | 0.6729 | 0.6675 | 0.5194 | 0.3984 | 0.4674 | 0.4651 | 0.4659 |
| | CAL [IJCAI 2025] | 0.5394 | 0.4606 | 0.2788 | 0.2182 | 0.1576 | 0.5567 | 0.4251 | 0.2593 | 0.2105 | 0.1100 | 0.2740 | 0.1193 | 0.0241 | 0.0092 | 0.0000 |
| | **CARD (Ours)** | **0.7636** | **0.7273** | **0.7152** | **0.7576** | **0.7030** | **0.7420** | 0.6868 | 0.6872 | **0.7328** | **0.7018** | **0.5521** | 0.4703 | 0.4750 | **0.5423** | **0.4682** |
| 100Leaves | PIC [IJCAI 2019] | 0.8369 | 0.7362 | 0.7181 | 0.6300 | 0.5112 | 0.9357 | 0.8516 | 0.8388 | 0.7814 | 0.7274 | 0.8026 | 0.6256 | 0.5955 | 0.4710 | 0.3496 |
| | EEIMVC [TPAMI 2021] | 0.8425 | 0.8081 | 0.7356 | 0.6694 | 0.5544 | 0.9214 | 0.8889 | 0.8494 | 0.8025 | 0.7357 | 0.7765 | 0.7113 | 0.6139 | 0.5147 | 0.3842 |
| | IMVC-CBG [CVPR 2022] | 0.6094 | 0.5525 | 0.4256 | 0.3869 | 0.2938 | 0.7783 | 0.6918 | 0.5986 | 0.5453 | 0.5408 | 0.3324 | 0.1742 | 0.1217 | 0.0920 | 0.1124 |
| | SIMVC-SA [MM 2023] | 0.6919 | 0.5881 | 0.4975 | 0.3887 | 0.3438 | 0.8333 | 0.7706 | 0.7106 | 0.6494 | 0.6266 | 0.5731 | 0.4448 | 0.3332 | 0.2149 | 0.1794 |
| | FCMVC-IVA [TIP 2024] | 0.4644 | 0.3956 | 0.3731 | 0.2919 | 0.2606 | 0.6959 | 0.6395 | 0.6138 | 0.5717 | 0.5445 | 0.3180 | 0.2318 | 0.2039 | 0.1379 | 0.1035 |
| | FIMVC-VIA [TNNLS 2024] | 0.7919 | 0.6994 | 0.6138 | 0.4738 | 0.3887 | 0.8926 | 0.8351 | 0.7696 | 0.6890 | 0.6530 | 0.7085 | 0.5776 | 0.4432 | 0.2891 | 0.2151 |
| | FSIMVC-OF [MM 2024] | 0.7662 | 0.7206 | 0.6744 | 0.6175 | 0.5837 | 0.8526 | 0.8134 | 0.8090 | 0.7515 | 0.7371 | 0.4106 | 0.3763 | 0.4262 | 0.3188 | 0.3010 |
| | LFIMVC [TIP 2024] | 0.8475 | 0.7837 | 0.7175 | 0.6338 | 0.5425 | 0.9251 | 0.8761 | 0.8323 | 0.7804 | 0.7257 | 0.7843 | 0.6777 | 0.5817 | 0.4746 | 0.3590 |
| | PSIMVC-PG [TNNLS 2024] | 0.4944 | 0.4750 | 0.4144 | 0.3425 | 0.3175 | 0.7312 | 0.7010 | 0.6547 | 0.5986 | 0.5829 | 0.3725 | 0.3223 | 0.2495 | 0.1719 | 0.1486 |
| | CAL [IJCAI 2025] | 0.1212 | 0.0938 | 0.0806 | 0.0519 | 0.0544 | 0.3450 | 0.2779 | 0.1931 | 0.1155 | 0.1005 | 0.0364 | 0.0232 | 0.0076 | 0.0017 | 0.0022 |
| | **CARD (Ours)** | **0.8763** | **0.8200** | **0.8031** | **0.7981** | **0.6394** | **0.9500** | **0.9090** | **0.8833** | **0.8547** | **0.7997** | **0.8450** | **0.7561** | **0.7162** | **0.6120** | **0.5090** |
| NUSWIDE | PIC [IJCAI 2019] | 0.1560 | 0.1560 | 0.1620 | 0.1525 | 0.1490 | 0.1795 | 0.1840 | 0.1790 | 0.1774 | 0.1595 | 0.0507 | 0.0491 | 0.0493 | 0.0515 | 0.0333 |
| | EEIMVC [TPAMI 2021] | 0.1575 | 0.1625 | 0.1600 | 0.1510 | 0.1435 | 0.1710 | 0.1762 | 0.1753 | 0.1590 | 0.1515 | 0.0465 | 0.0453 | 0.0436 | 0.0420 | 0.0371 |
| | IMVC-CBG [CVPR 2022] | 0.1605 | 0.1595 | 0.1590 | 0.1640 | 0.1570 | 0.1401 | 0.0722 | 0.0569 | 0.1114 | 0.0769 | 0.0271 | 0.0078 | 0.0025 | 0.0206 | 0.0178 |
| | SIMVC-SA [MM 2023] | 0.1370 | 0.1345 | 0.1430 | 0.1350 | 0.1375 | 0.1600 | 0.1484 | 0.1450 | 0.1437 | 0.1441 | 0.0333 | 0.0270 | 0.0323 | 0.0316 | 0.0300 |
| | FCMVC-IVA [TIP 2024] | 0.1275 | 0.1240 | 0.1155 | 0.1155 | 0.1035 | 0.1433 | 0.1331 | 0.1201 | 0.1189 | 0.1033 | 0.0310 | 0.0234 | 0.0192 | 0.0190 | 0.0119 |
| | FIMVC-VIA [TNNLS 2024] | 0.1495 | 0.1495 | 0.1485 | 0.1490 | 0.1430 | 0.1732 | 0.1551 | 0.1536 | 0.1513 | 0.1512 | 0.0418 | 0.0376 | 0.0360 | 0.0395 | 0.0342 |
| | FSIMVC-OF [MM 2024] | 0.1570 | 0.1435 | 0.1380 | 0.1465 | 0.1360 | 0.1674 | 0.1652 | 0.1479 | 0.1445 | 0.1454 | 0.0476 | 0.0328 | 0.0265 | 0.0266 | 0.0207 |
| | LFIMVC [TIP 2024] | 0.1425 | 0.1365 | 0.1390 | 0.1365 | 0.1185 | 0.1571 | 0.1440 | 0.1479 | 0.1373 | 0.1284 | 0.0356 | 0.0279 | 0.0334 | 0.0289 | 0.0192 |
| | PSIMVC-PG [TNNLS 2024] | 0.1345 | 0.1370 | 0.1305 | 0.1300 | 0.1320 | 0.1588 | 0.1584 | 0.1487 | 0.1489 | 0.1504 | 0.0314 | 0.0328 | 0.0288 | 0.0321 | 0.0275 |
| | CAL [IJCAI 2025] | 0.1245 | 0.1250 | 0.1170 | 0.1245 | 0.1220 | 0.0166 | 0.0252 | 0.0230 | 0.0162 | 0.0167 | 0.0005 | 0.0005 | 0.0007 | 0.0001 | 0.0002 |
| | **CARD (Ours)** | **0.1625** | **0.1635** | **0.1665** | **0.1760** | **0.1650** | **0.1834** | **0.1919** | **0.1844** | **0.1866** | **0.1742** | **0.0510** | **0.0515** | **0.0515** | **0.0585** | **0.0441** |

datasets under missing ratios $\rho \in \{0.1, 0.3, 0.5, 0.7, 0.9\}$.

Across the tested datasets and missing settings, CARD generally shows stable behavior over the searched hyperparameter grid. In most cases, the ACC surfaces remain smooth and relatively flat over a broad range of anchor numbers and diffusion-related parameters, indicating that the proposed method is not overly sensitive to hyperparameter choices within the tested range.

Figures 3 and 4 summarize the parameter sensitivity results on the retained benchmark datasets, covering a wide range of data scales, view numbers, and semantic complexities. For each dataset, performance surfaces are reported under five missing ratios to assess robustness under different degrees of incompleteness.

On small- and medium-scale datasets such as Yale, 100Leaves, NUSWIDE, and UCIDigits, CARD generally exhibits low sensitivity to parameter variations, and similar trends are observed across different missing ratios. On more challenging datasets, including Scene-15, CCV, SUNRGBD, and NUSWIDEOBJ, the method remains relatively stable over a wide parameter range, with most variation occurring only under very small anchor budgets at high missing ratios.

Overall, these results suggest that CARD is numerically stable and not overly sensitive with respect to hyperparameter choices across datasets of different scales and under varying degrees of incompleteness. Together with the representative results reported in the main text, this additional analysis further supports the practical stability of the proposed method.

### C.7. Additional Convergence Study

This subsection provides a comprehensive convergence analysis to support the observations reported in the main text. We report convergence curves of the objective function in Eq. (2) on all benchmark datasets under missing ratios $\rho \in \{0.1, 0.3, 0.5, 0.7, 0.9\}$.

Across all datasets and missing settings, CARD exhibits consistent and stable convergence behavior. In all cases, the objective value decreases rapidly during the early iterations and quickly stabilizes after a small number of outer iterations. No evident oscillation or divergence is observed, even under severe missing-view scenarios with $\rho = 0.9$, empirically supporting the numerical stability of the proposed refinement procedure in the tested settings.

*Table 5.* Clustering performance under complete settings (missing rate $\rho = 0$). "–" indicates that the method was not runnable under the same computational budget due to memory or time limitations.

| Method | Metric | Yale | 100Leaves | NUSWIDE | UCIDigits | Scene-15 | CCV | SUNRGBD | NUSWIDEOBJ |
|---|---|---|---|---|---|---|---|---|---|
| EOMSC-CA [AAAI 2022] | ACC | 0.6485 | 0.5094 | 0.2105 | 0.6635 | 0.3380 | 0.2432 | **0.2370** | **0.1981** |
| | NMI | 0.6541 | 0.7129 | 0.1775 | 0.7320 | 0.3062 | 0.1859 | 0.2203 | 0.1221 |
| | ARI | 0.4296 | 0.2262 | **0.0690** | 0.6097 | 0.1868 | 0.0852 | 0.0966 | 0.0669 |
| FSSC [TNNLS 2022] | ACC | 0.4727 | 0.7413 | 0.1725 | 0.5160 | 0.3010 | 0.2098 | 0.1509 | 0.1649 |
| | NMI | 0.5075 | 0.8796 | 0.1647 | 0.5123 | 0.2785 | 0.1762 | 0.0414 | 0.1287 |
| | ARI | 0.2646 | 0.6511 | 0.0506 | 0.3772 | 0.1684 | 0.0744 | 0.0204 | 0.0513 |
| FPMVS-CAG [TIP 2022] | ACC | 0.4364 | 0.3150 | 0.2070 | 0.7310 | 0.3338 | 0.2255 | 0.2358 | 0.1913 |
| | NMI | 0.4906 | 0.6232 | 0.1761 | 0.7053 | 0.3126 | 0.1491 | 0.2315 | 0.1186 |
| | ARI | 0.2479 | 0.2024 | 0.0588 | 0.6035 | 0.1833 | 0.0702 | 0.0963 | 0.0632 |
| FCAG [TPAMI 2024] | ACC | 0.2485 | 0.0013 | 0.0210 | 0.4120 | 0.1369 | 0.1221 | 0.2151 | 0.0167 |
| | NMI | 0.2496 | 0.8195 | 0.0732 | 0.4112 | 0.0678 | 0.0332 | 0.1648 | 0.0847 |
| | ARI | 0.0525 | 0.4757 | 0.0072 | 0.3108 | 0.0108 | 0.0016 | 0.0775 | 0.0332 |
| UDBGL [TNNLS 2024] | ACC | 0.2909 | 0.1656 | 0.1380 | 0.2385 | 0.3659 | 0.1426 | 0.1328 | 0.1332 |
| | NMI | 0.2959 | 0.2135 | 0.1627 | 0.2188 | 0.3931 | 0.0987 | 0.0244 | 0.1052 |
| | ARI | 0.0630 | 0.0051 | 0.0214 | 0.0751 | 0.2233 | 0.0176 | 0.0000 | 0.0373 |
| EDCAG [TNNLS 2024] | ACC | 0.5030 | 0.7763 | 0.1615 | 0.7275 | 0.3347 | 0.1866 | 0.1535 | 0.1583 |
| | NMI | 0.5647 | 0.9154 | 0.1885 | 0.6856 | 0.3481 | 0.1669 | 0.1908 | 0.1375 |
| | ARI | 0.3318 | 0.7322 | 0.0522 | 0.5818 | 0.1770 | 0.0586 | 0.0690 | 0.0576 |
| BGAE [TKDE 2024] | ACC | 0.6909 | 0.4856 | 0.1825 | 0.4775 | 0.3186 | 0.2182 | 0.2342 | 0.1659 |
| | NMI | 0.7231 | 0.7123 | 0.1689 | 0.5050 | 0.3132 | 0.1792 | 0.2350 | 0.1192 |
| | ARI | 0.5367 | 0.2297 | 0.0570 | 0.4002 | 0.1932 | 0.0775 | 0.0953 | 0.0590 |
| AEVC [CVPR 2024] | ACC | 0.5515 | 0.6025 | 0.1635 | 0.8140 | 0.3906 | 0.2247 | 0.1606 | 0.1393 |
| | NMI | 0.6480 | 0.7582 | 0.1775 | 0.7483 | 0.3782 | 0.1705 | 0.1878 | 0.1343 |
| | ARI | 0.4169 | 0.1575 | 0.0375 | 0.6480 | 0.2088 | 0.0668 | 0.0651 | 0.0460 |
| ALPC [AAAI 2025] | ACC | 0.5879 | 0.0144 | 0.1665 | 0.7365 | 0.4033 | 0.2243 | 0.1724 | 0.1403 |
| | NMI | 0.6105 | 0.0745 | 0.1904 | 0.6483 | 0.4203 | 0.1803 | 0.2132 | 0.1233 |
| | ARI | 0.3847 | 0.0012 | 0.0509 | 0.5481 | 0.2531 | 0.0704 | 0.0755 | 0.0428 |
| SSA-MVC [NeurIPS 2025] | ACC | 0.7212 | 0.8531 | 0.1565 | 0.9385 | 0.4615 | 0.2036 | – | – |
| | NMI | 0.7242 | 0.9348 | 0.1693 | 0.8820 | 0.4375 | 0.1656 | – | – |
| | ARI | 0.5398 | 0.8078 | 0.0446 | 0.8806 | 0.2768 | 0.0653 | – | – |
| **CARD (Ours)** | ACC | **0.7712** | **0.8925** | 0.1725 | **0.9670** | **0.4908** | **0.2984** | 0.2348 | 0.1759 |
| | NMI | **0.7485** | **0.9601** | **0.1986** | **0.9028** | **0.4646** | **0.2173** | **0.2451** | **0.1504** |
| | ARI | **0.5698** | **0.8691** | 0.0619 | **0.9201** | **0.3276** | **0.1117** | **0.1131** | **0.0709** |

Figures 5 and 6 summarize the convergence behavior on eight benchmark datasets with diverse characteristics, including variations in data scale, number of views, and semantic complexity. For each dataset, convergence curves are reported under five missing ratios to verify the stability of CARD across different degrees of incompleteness.

On small- and medium-scale datasets such as Yale, 100Leaves, NUSWIDE, UCIDigits, and Scene-15, CARD often stabilizes within a few outer iterations, and this rapid stabilization is largely insensitive to the missing ratio. On larger and more challenging datasets, including CCV, SUNRGBD, and NUSWIDEOBJ, the objective value also decreases steadily and reaches a stable solution without noticeable fluctuations, despite the increased data scale and structural complexity.

Overall, these results provide empirical evidence that CARD exhibits stable numerical behavior in the tested settings, complementing the convergence discussion in the main paper.

## C.8. Details of the Ablation Study

This subsection provides detailed formulations and implementation details of the ablation variants reported in Table 2 of the main text. All ablation experiments are conducted on complete data ($\rho = 0$) to ensure a controlled and interpretable

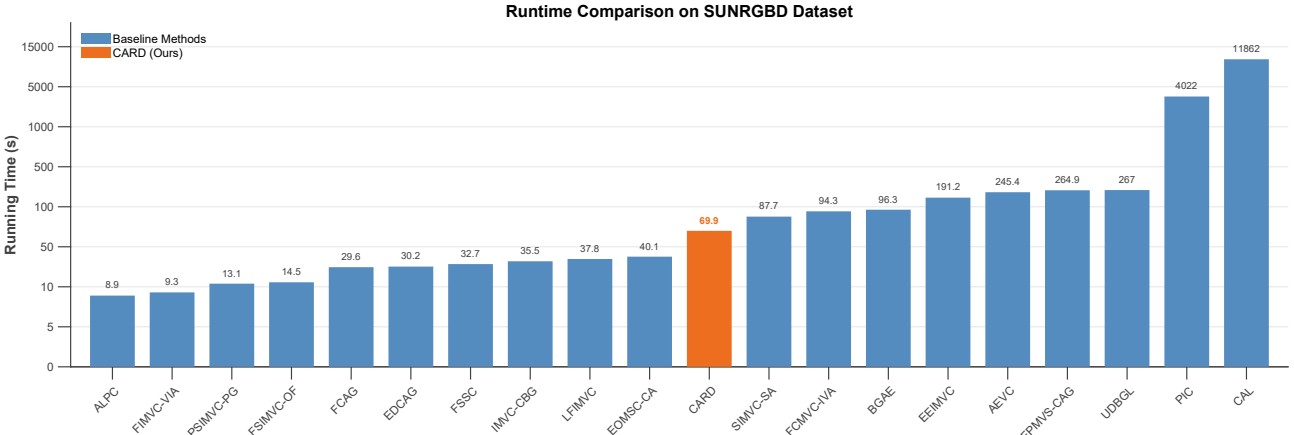

*Figure 2.* Single-run runtime comparison (in seconds) on the SUNRGBD dataset, excluding hyperparameter search. CARD shows competitive efficiency. SSA-MVC is omitted because it triggered an out-of-memory error in our environment.

comparison, where performance differences can be attributed to architectural components rather than to missing-view effects.

The ablation variants are designed to examine two key components of CARD: (i) high-order resolvent diffusion on the unified anchor hypergraph, and (ii) clustering feedback through the $\mathbf{C} \leftrightarrow \mathbf{Y}$ coupling. We describe each variant in turn below.

**Avg. Anchor Kernel.** This variant removes both high-order resolvent diffusion and clustering feedback. Each view independently constructs its anchor-induced kernel, and the view-specific kernels are averaged:

$$P_{\text{intra}} \;=\; \frac{1}{V} \sum_{v=1}^{V} \mathbf{Q}_v \mathbf{Q}_v^{\top}, \tag{74}$$

where $\mathbf{Q}_v$ denotes the normalized view-specific sample–anchor incidence matrix constructed in the same way as $\mathbf{Q}$ but using only view $v$. This variant removes cross-view anchor coupling while keeping the same normalization principle. Clustering assignments are then obtained by directly applying kernel K-means on $P_{\text{intra}}$.

**+Resolvent Diffusion.** To examine the effect of high-order diffusion without clustering feedback, we retain the unified anchor diffusion operator used by CARD while keeping the anchor-kernel target fixed. Specifically, this variant solves

$$\min_{\mathbf{C}} \; \text{Tr}\big(\mathbf{C}^{\top}\mathbf{C} - \mathbf{C}^{\top}\mathbf{A}\mathbf{C}\mathbf{A}^{\top}\big) \;+\; \mu\|\mathbf{C} - P_{\text{intra}}\|_F^2, \tag{75}$$

where $\mathbf{A} = \mathbf{Q}\mathbf{Q}^{\top}$ is the normalized unified anchor-hypergraph diffusion operator constructed from the concatenated multi-view anchor incidence matrix, and $P_{\text{intra}}$ is the fixed anchor-kernel target defined above. After convergence, clustering assignments are obtained by applying kernel K-means to $\mathbf{C}$, without feeding the resulting partition back to update $\mathbf{C}$.

**CARD.** Finally, the full CARD model replaces the fixed anchor-kernel target with the cluster-induced projector $\mathbf{S}(\mathbf{Y})$ and performs the self-consistency refinement described in Section 3.5 of the main paper. This $\mathbf{C} \leftrightarrow \mathbf{Y}$ feedback enables affinity learning and clustering to mutually refine each other.

Overall, these ablation variants progressively introduce high-order diffusion and clustering feedback under a controlled setting. This formulation allows Table 2 to reflect the contribution of resolvent diffusion and the $\mathbf{C} \leftrightarrow \mathbf{Y}$ coupling without relying on any one-to-one correspondence assumption among view-specific anchors.

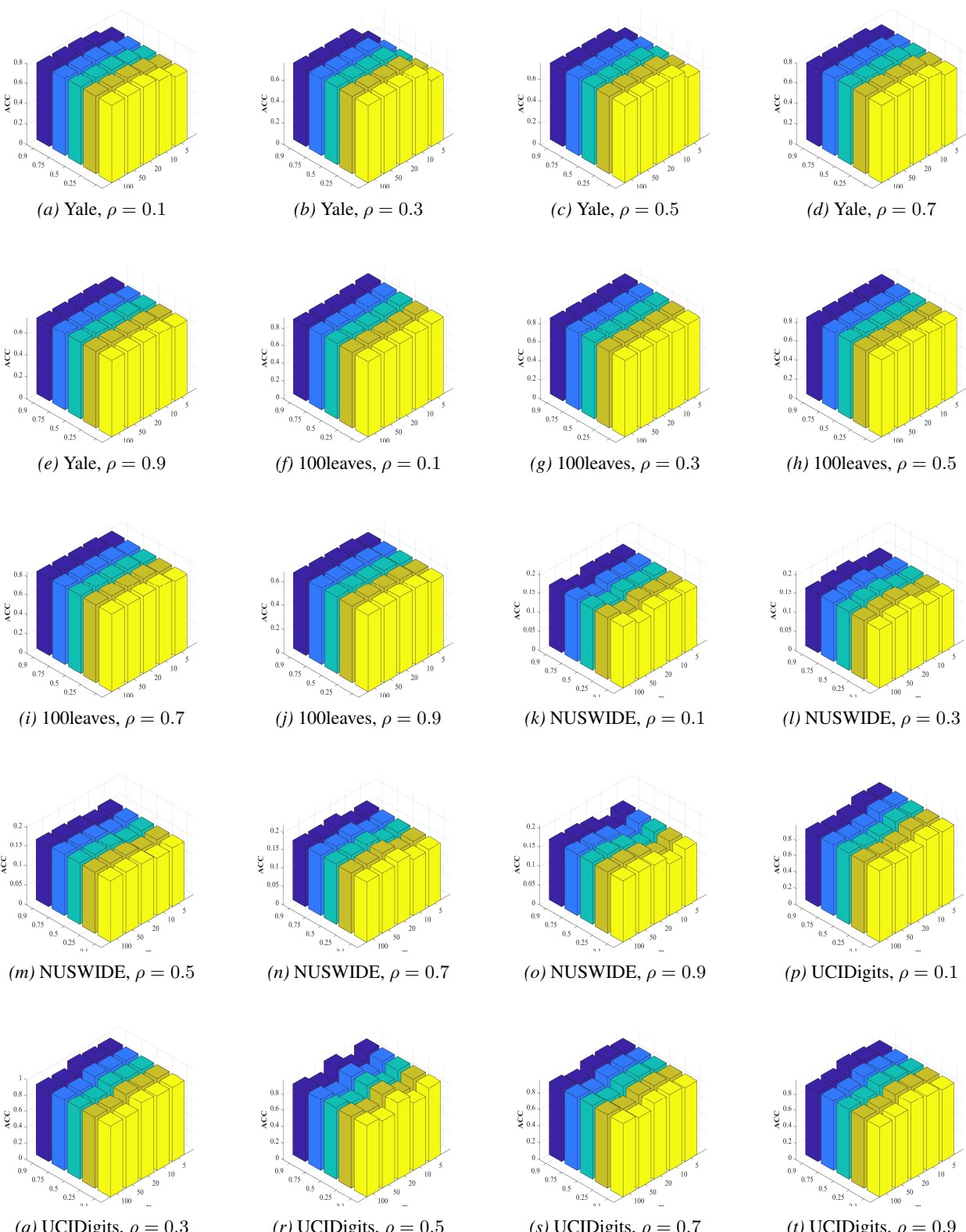

*Figure 3.* Parameter sensitivity analysis on **Yale**, **100leaves**, **NUSWIDE**, and **UCI Digits** datasets.

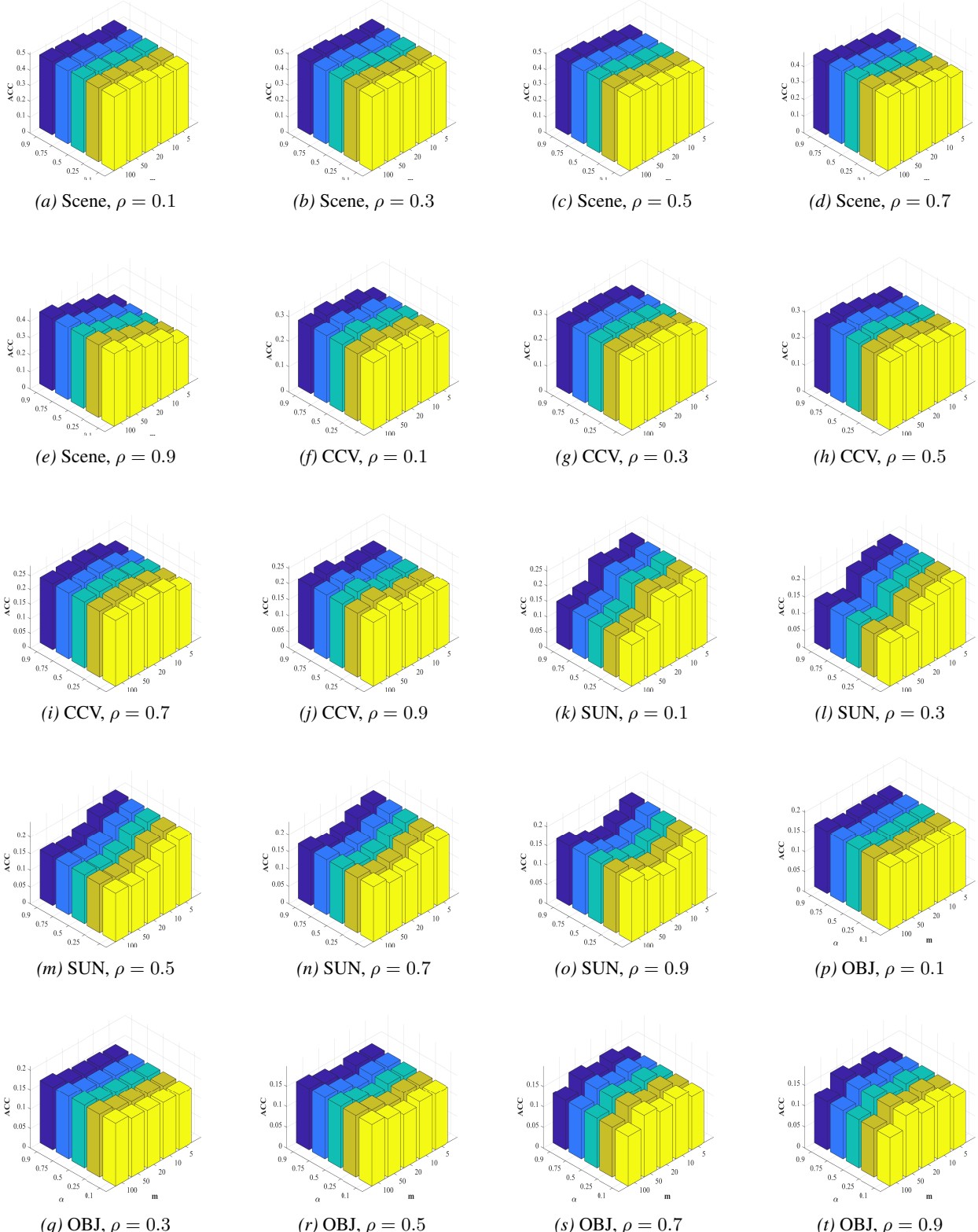

*Figure 4.* Parameter sensitivity analysis on **Scene-15**, **CCV**, **SUNRGBD**, and **NUSWIDEOBJ** datasets.

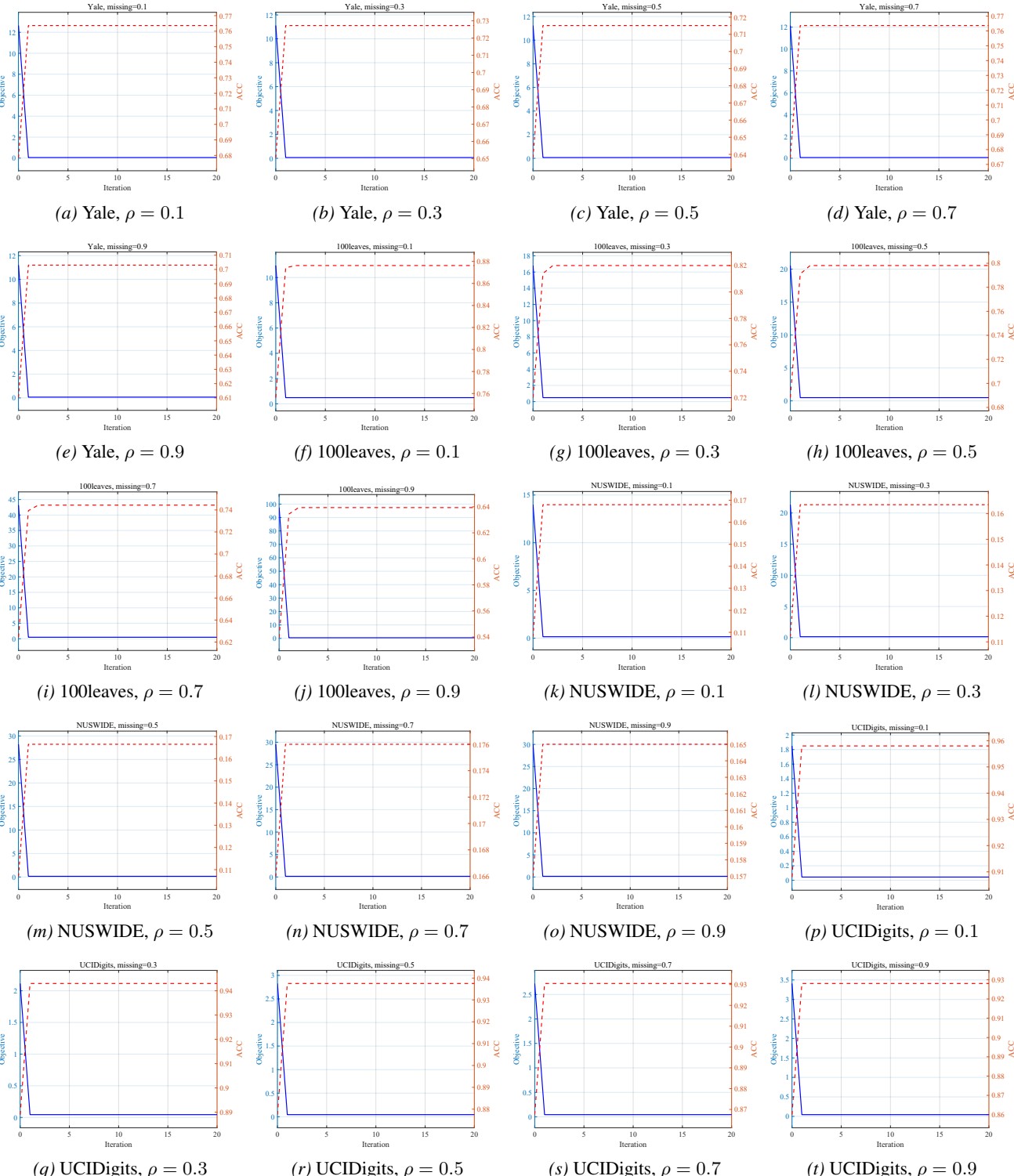

*Figure 5.* Convergence analysis on **Yale**, **100leaves**, **NUSWIDE**, and **UCI Digits** datasets.

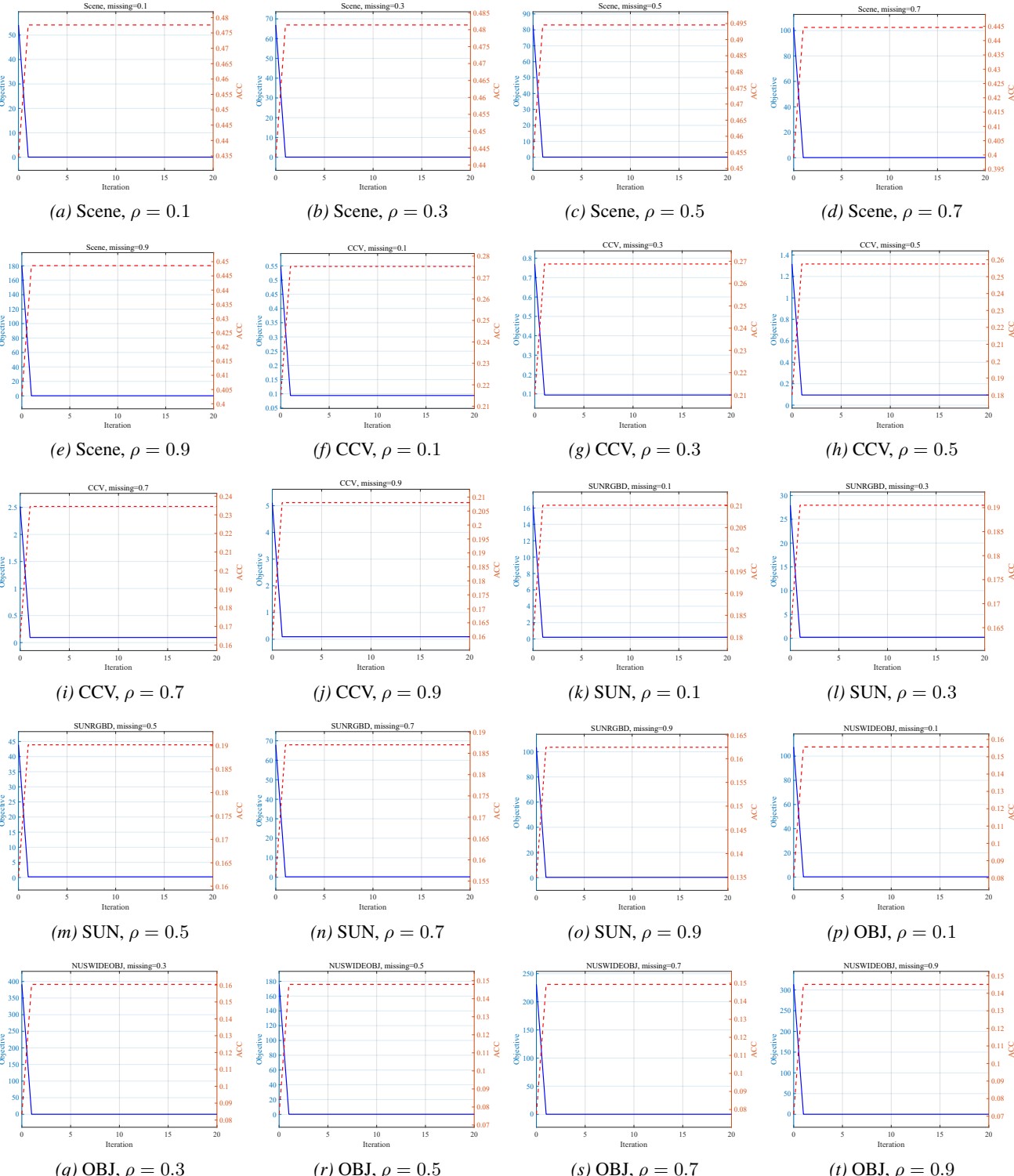

*Figure 6.* Convergence analysis on **Scene-15**, **CCV**, **SUNRGBD**, and **NUSWIDEOBJ** datasets.

