# OpenReview forum: "Contractive Anchor Resolvent Diffusion for Incomplete Multi-View Clustering"
_ICML.cc/2026/Conference — ICML 2026 regular_

### Official Review · Reviewer_QNg7 · 2026-03-10

**Soundness:** 3
**Presentation:** 3
**Significance:** 3
**Originality:** 3
**Overall Recommendation:** 4
**Confidence:** 5

**Summary:**

This paper tackles Incomplete Multi-View Clustering (IMVC) by identifying structural degradation of similarity graphs as the core bottleneck. It proposes CARD, a novel framework that formulates IMVC as high-order spectral filtering via an anchor-induced hypergraph and resolvent diffusion, avoiding explicit data imputation while enhancing consensus structures and reducing view-specific noise. It also provides an implicit linear-time solver with theoretical guarantees and state-of-the-art performance.

**Compliance With Llm Reviewing Policy:**

Affirmed.

**Final Justification:**

My concerns have now been resolved, and I maintain my positive rating.

**Key Questions For Authors:**

1. Propositions 4.2 and 4.5 highlight a critical theoretical trade-off: a larger $\alpha$ improves spectral selectivity against noise but exponentially degrades the solver's condition number. Could the authors explicitly visualize this phase transition boundary via a controlled, detailed simulation? Demonstrating the exact critical point where information-theoretic gains (spectral filtering) are overtaken by numerical instability (solver crash) would deeply enrich the analysis.
2. Extreme missing rates $\rho$ essentially simulate a percolation process that eventually shatters the graph's global skeleton.  Rescuing global connectivity near this percolation threshold requires pushing $\alpha \to 1^-$ to activate infinite-length paths, which inherently risks numerical collapse. Could the authors provide fine-grained empirical results across varying $\rho$ and $\alpha$? This would clarify the true operational bottleneck of the CARD framework under extreme incompleteness: does the framework ultimately fail due to a numerical explosion (when forcing $\alpha \to 1$), or because of a true topological disconnection (the graph physically shatters)?

**Limitations:**

Yes.

**Strengths And Weaknesses:**

1. Shifting the IMVC paradigm away from traditional feature imputation toward high-order spectral filtering of structural evidence is a highly creative and effective perspective. The Graph Signal Processing (GSP) formulation for IMVC, coupled with the exact implicit reduction to dynamic Euclidean K-means, represents a distinct and innovative contribution to the field.
2. The mathematical foundation is well-motivated. Translating high-order relational consistency energy into a resolvent operator is a highly principled approach to handling structural noise. The proofs provided in the appendix are meticulous, theoretically sound, and build cohesively upon established mathematical tools.
3. The paper tackles a major bottleneck in multi-view learning: scaling robust clustering to large, severely incomplete datasets. Achieving true linear complexity while fully utilizing high-order manifold structures is a significant advancement for large-scale optimization, bridging the gap between graph learning theory and scalable clustering practice.

While the paper is clearly written structurally, the main text is excessively dense mathematically. The heavy notation creates a barrier, making it difficult for readers to extract the key high-level intuitions driving the framework. To maximize its impact and make the methodology accessible to a broader machine learning audience, a subsequent revision should heavily prioritize clarity and intuitive, high-level explanations alongside the equations.

---

> ### Author Rebuttal · Authors · 2026-03-30
>
> W1:We agree, and in the revision we will improve clarity by adding more intuitive high-level explanations in the main text and reducing the burden of dense notation where possible.
>
> Q1：We have added a controlled $(p,\alpha)$ sweep; see the phase diagram at [alpha_failure_boundary_phase_diagram.png](https://anonymous.4open.science/r/CARD-8CB1/Rebuttal/alpha_failure_boundary_phase_diagram.png) and the detailed table at [alpha_failure_boundary_summary.csv](https://anonymous.4open.science/r/CARD-8CB1/Rebuttal/alpha_failure_boundary_summary.csv). Here, the numerical-risk boundary is tracked by the collapse of the resolvent margin $\min_{i,j}(1-\alpha\lambda_i\lambda_j)$. Empirically, we observe a clear numerical-risk boundary as $\alpha\to1^{-}$: on both SUNRGBD and NUSWIDEOBJ, the numerical boundary starts to appear around $\alpha\approx0.999$ to $0.9995$, while ACC no longer improves. Thus, we can identify an empirical risk boundary where the practical benefit of pushing $\alpha$ further essentially disappears. We do not claim that this is yet an exact solver-crash point, since actual solver failure is not observed in the tested range, but it already makes the trade-off in Propositions 4.2 and 4.5 visible in practice.
>
> Q2：We also use the same $(p,\alpha)$ sweep to clarify the operational bottleneck under extreme incompleteness; see [alpha_failure_boundary_phase_diagram.png](https://anonymous.4open.science/r/CARD-8CB1/Rebuttal/alpha_failure_boundary_phase_diagram.png) and [alpha_failure_boundary_summary.csv](https://anonymous.4open.science/r/CARD-8CB1/Rebuttal/alpha_failure_boundary_summary.csv). On the structural side, we monitor the cluster-boundary spectral gap $\lambda_c-\lambda_{c+1}$ and the largest-component ratio of the anchor-induced graph. The results suggest that the failure mode is dataset-dependent but clearly structured. On SUNRGBD, after entering the numerical-risk region, the graph also becomes visibly fragmented under the most extreme setting $(\rho=0.99,\alpha=0.9999)$, where the largest-component ratio drops to about $0.50$. On NUSWIDEOBJ, by contrast, the largest-component ratio remains near $1$, while the spectral gap stays extremely small, indicating a collapse of discriminative graph structure rather than a literal physical disconnection. Therefore, under extreme incompleteness, CARD can fail either through numerical risk or through structural degeneration, but in our experiments the dominant mechanism is not a sudden solver crash; it is the progressive loss of useful graph structure as $\alpha$ is pushed too close to $1$.

---

> > ### Author Rebuttal · Reviewer_QNg7 · 2026-04-02
> >
> > Thanks for the authors‘ response. My concerns have now been resolved.

---

### Official Review · Reviewer_GvAn · 2026-03-10

**Soundness:** 3
**Presentation:** 4
**Significance:** 4
**Originality:** 4
**Overall Recommendation:** 5
**Confidence:** 4

**Summary:**

This paper studies incomplete multi-view clustering (IMVC) and proposes CARD (Contractive Anchor Resolvent Diffusion), a scalable graph/hypergraph-based framework that avoids explicit data imputation. The core idea is to reinterpret IMVC as a spectral filtering problem under structural degradation caused by missing views. The method constructs a unified anchor-induced hypergraph over observed relations and introduces a high-order resolvent diffusion operator to enhance consensus structure while suppressing view-specific noise. On the optimization side, the paper derives a closed-form characterization of the similarity matrix and further reformulates the alternating optimization into an explicit feature construction followed by Euclidean K-means, yielding linear complexity in the number of samples. The paper also provides several theoretical claims, including spectral selectivity, spectral mass recovery, local contraction behavior, conditioning analysis, and robustness to anchor perturbation. Empirically, the method shows strong results across a broad range of incomplete-view clustering benchmarks.

**Compliance With Llm Reviewing Policy:**

Affirmed.

**Final Justification:**

I confirm my score for paper acceptance.

**Key Questions For Authors:**

1. Could the authors provide an ablation study under incomplete settings (e.g., (\rho=0.5, 0.7, 0.9)) rather than only on complete data?
2. It would be more convincing to visualize eigenvalue distributions, eigengap changes, or related spectral statistics before and after resolvent diffusion.
3. Since the method relies heavily on anchor-induced structure, it would strengthen the paper to report sensitivity across different random seeds, anchor sets, or alternative anchor construction strategies.

**Limitations:**

Yes

**Strengths And Weaknesses:**

Strengths
1. A fresh perspective on IMVC. Framing missing-view clustering as a spectral distortion / structural inference problem rather than a feature imputation problem is conceptually appealing and helps differentiate the work from both generative completion methods and standard graph fusion methods.

2. Technically interesting diffusion formulation. The resolvent-based diffusion operator is more expressive than first-order fusion, and the compact low-rank derivation through the anchor subspace is elegant. The kernel-to-Euclidean reformulation is also useful from a practical standpoint.

Weaknesses
1. Theoretical claims seem stronger than what is fully justified in the main text.
   Several theoretical statements are appealing, but currently read more like heuristic or semi-formal arguments than fully rigorous guarantees. In particular:
    (1)The “local contraction mapping toward the consensus subspace” result depends on a basin-of-attraction assumption, which weakens the practical interpretability of the guarantee.
(2)Proposition 1 on “asymptotic spectral separation” appears oversimplified. The comparison with “any first-order linear fusion operator” is too broad unless the class of operators is explicitly formalized.

2. The ablation is informative, but it is conducted only on complete data. Since the paper’s main motivation is specifically incomplete multi-view clustering under severe missingness, it would be stronger to include at least one ablation under nontrivial missing ratios (e.g., $\rho=0.5$ and $\rho=0.9$).

---

> ### Author Rebuttal · Authors · 2026-03-30
>
> W1：We agree that these theoretical statements should be scoped more carefully, and we will revise them accordingly.
>
> W2、Q1：We have now added ablation results under nontrivial missingness. For the first-order variant, on SUNRGBD the performance is 0.1787/0.1311/0.0153 at $\rho=0.5$ and 0.1536/0.1403/0.0251 at $\rho=0.9$ in ACC/NMI/ARI. On NUSWIDEOBJ, it is 0.1517/0.1038/0.0190 at $\rho=0.5$ and 0.1385/0.0847/0.0076 at $\rho=0.9$. We will include these incomplete-setting ablations in the revision.
>
> Q2：We have added a real-data spectral-gap visualization on SUNRGBD and NUSWIDEOBJ; see [spectral_gap_visualization](https://anonymous.4open.science/r/CARD-8CB1/Rebuttal/spectral_gap_visualization.png). The plot shows that the empirical cluster-boundary gap $\lambda_c-\lambda_{c+1}$ remains nonzero across the tested missing ratios, which directly supplements the operator-level discussion in the paper. We also provide a toy spectral visualization to illustrate the before/after filtering effect more directly: [toy_spectral_visualization](https://anonymous.4open.science/r/CARD-8CB1/Rebuttal/toy_spectral_visualization.png)
>
> Q3:We have added anchor-robustness experiments under severe missingness by comparing K-means anchors and random anchors across multiple seeds. The result consistently favors K-means anchors: for example, on SUNRGBD, ACC drops from 0.1870 / 0.1624 to 0.1338 / 0.1294 at $\rho=0.7$ and $\rho=0.9$; on NUSWIDEOBJ, it drops from 0.1560 / 0.1542 to 0.1159 / 0.1291. This shows that anchor quality matters, while the method remains reasonably stable under standard anchor perturbations.

---

> > ### Author Rebuttal · Reviewer_GvAn · 2026-04-01
> >
> > I thanks the authors for the clarifications. I confirm my score for paper acceptance.

---

### Official Review · Reviewer_6HSE · 2026-03-12

**Soundness:** 3
**Presentation:** 4
**Significance:** 3
**Originality:** 4
**Overall Recommendation:** 4
**Confidence:** 5

**Summary:**

The paper introduces Contractive Anchor Resolvent Diffusion (CARD), a framework for Incomplete Multi-View Clustering (IMVC) that avoids data imputation by reformulating the problem as high-order spectral filtering on a unified anchor-induced hypergraph. It derives a resolvent diffusion operator to enhance consensus structures while suppressing noise from missing views, couples it with a joint optimization for similarity learning and clustering via an implicit solver, and achieves linear complexity. Theoretical analysis includes spectral selectivity, mass recovery bounds, and local contraction guarantees. Experiments on benchmark datasets under varying missing rates show consistent superiority over baselines, with code available for reproduction.

**Compliance With Llm Reviewing Policy:**

Affirmed.

**Key Questions For Authors:**

1. Experiments use random uniform missing; how does CARD perform under structured missing (e.g., class-correlated missing)?
2. Anchor selection is K-means on observed samples; why not learned anchors or view-adaptive counts?
3. Ablation on alternatives might strengthen claims, as poor anchors could degrade the hypergraph.
4. $\mu$ controls diffusion; is there an adaptive selection? Manual tuning in supplements works, but automation could enhance practicality and significance.

**Strengths And Weaknesses:**

Strengths
1. The technical claims are well-supported. The reformulation of IMVC as spectral filtering is principled, with the resolvent operator correctly derived from the Sylvester equation. Theoretical guarantees regarding spectral selectivity, mass recovery, and local contraction are solid, and the assumptions are perfectly reasonable for clustering tasks. The experimental design follows rigorous, commonly used protocols, featuring benchmarks across varying missing rates, comprehensive ablation studies, parameter sensitivity analysis, convergence tracking, and runtime comparisons.
2. The paper introduces highly novel elements to the field. Reformulating IMVC as high-order resolvent diffusion on anchor hypergraphs, deriving an implicit solver for joint optimization, and providing explicit contraction guarantees are highly creative steps. While building upon Graph Signal Processing (GSP) and anchor graphs, the dynamic K-means reformulation and high-order operator inject fresh insights and inspiration into the domain.
3. This work tackles a critical bottleneck in IMVC, structural degradation caused by missing views, through a unique spectral perspective. This approach has a strong potential to influence scalable clustering practices in large-scale, incomplete data scenarios. Furthermore, the proposed resolvent operator and implicit solver show great promise for generalization to other graph-based tasks beyond IMVC.

Weaknesses
Although I did not identify any obvious mathematical flaws during my review, the sheer complexity warrants a rigorous second pass by the authors. I strongly advise the authors to carefully proofread and cross-check all equations, symbol definitions, theoretical assumptions, and proofs for absolute accuracy and consistency.

Please ensure that all assumptions are strictly valid and not over-claimed. Furthermore, streamlining the notation and providing more intuitive, high-level explanations alongside the dense formulas would not only preempt any potential rigorousness issues but also vastly improve the paper's accessibility to a broader audience.

---

> ### Author Rebuttal · Authors · 2026-03-30
>
> W1：We agree that the current presentation is denser than necessary. In the revision, we will carefully re-check notation, assumptions, and theorem statements; narrow the language where needed to avoid overclaiming; add high-level explanations around the key equations; keep the main text focused on intuition and move lower-level derivations to proof sketches / appendix.
>
> Q1: Beyond the standard benchmark protocol, we additionally tested a class-dependent missing mechanism in which missingness is concentrated on head classes. This setting is intended as a stress test rather than a replacement for the standard random-missing benchmark. Averaged over 5 missing-pattern seeds, CARD remains workable on both SUNRGBD and NUSWIDEOBJ under this structured shift, indicating that the framework is not restricted to a single missingness protocol.
>
> Q2：Our goal in this paper is to demonstrate the effectiveness of the CARD framework itself. The specific way of constructing the anchor-based bipartite graph is not the main focus of this work, but rather one reasonable and standard design choice. We use K-means anchors on observed samples because it is simple, stable, and widely adopted in anchor-based clustering. The key point is that CARD operates on the resulting anchor graph, so the proposed high-order diffusion mechanism is not tied to this particular anchor-selection rule. In this sense, we believe the framework is generic and can also be combined with learned anchors or view-adaptive anchor counts, which we view as natural extensions rather than core ingredients of the current paper.
>
> Q3：We agree that poor anchors can weaken the anchor graph. To check this directly, we compared K-means anchors with random anchors under severe missingness. The result consistently favors K-means anchors: for example, on SUNRGBD, ACC drops from 0.1870 / 0.1624 to 0.1338 / 0.1294 at $\rho=0.7$ and $\rho=0.9$; on NUSWIDEOBJ, it drops from 0.1560 / 0.1542 to 0.1159 / 0.1291. This confirms that anchor quality matters, while CARD itself is not tied to a specific anchor-learning mechanism.
>
> Q4：In this work, we select $\mu$ by grid search, which is the same protocol commonly used in prior IMVC methods and therefore provides a fair comparison. Our current goal is to evaluate CARD under the standard experimental setting rather than introduce an additional model-selection component. That said, adaptive selection is interesting, and in future work we plan to explore choosing $\mu$ via internal clustering criteria, e.g. the Silhouette Index (SI) or related unsupervised indicators.

---

> > ### Author Rebuttal · Reviewer_6HSE · 2026-04-01
> >
> > The authors addressed my concerns, so I maintained my score.

---

### Official Review · Reviewer_Lgt8 · 2026-03-22

**Soundness:** 4
**Presentation:** 3
**Significance:** 4
**Originality:** 4
**Overall Recommendation:** 5
**Confidence:** 4

**Summary:**

The paper introduces CARD, a framework for incomplete multi-view clustering (IMVC) that formulates the core problem as spectral distortion caused by missing views rather than missing feature values. Instead of imputation or first-order linear fusion, the authors build a unified anchor-induced hypergraph and introduce a high-order resolvent diffusion operator that acts as a sharp rational filter. They derive an implicit solver that jointly optimizes the similarity matrix and cluster assignments without ever materializing dense matrices, prove that the iteration is a local contraction mapping toward the consensus subspace, and deliver linear-time complexity. Experiments show good results under various missing rates.

This is a solid, well-motivated contribution that moves IMVC beyond the imputation/linear-fusion paradigm with both theoretical guarantees and practical scalability. The weaknesses are mostly presentation-related and do not undermine the core technical novelty or empirical strength.

**Compliance With Llm Reviewing Policy:**

Affirmed.

**Final Justification:**

This is a solid, well-motivated contribution that moves IMVC beyond the imputation/linear-fusion paradigm with both theoretical guarantees and practical scalability. The author has addressed the concerns raised through detailed responses. I believe that the paper exhibits significant advantages in terms of innovation, completeness, and logicality. I recommend accepting it.

**Key Questions For Authors:**

1. $\alpha$ tuning: The grid {0.1, …, 0.9} is used, yet Theorem 4 shows $\alpha$ strongly affects the contraction rate. In practice, does a fixed $\alpha \approx 0.7$ work across most datasets, or does the optimal value shift noticeably with missing rate? A short empirical rule or sensitivity plot would be useful.
2. Anchor selection under extreme incompleteness: When some samples have only one observed view, K-means anchors are learned only on observed data. Does this exacerbate anchor shift? A short ablation on anchor quality (e.g., random vs. k-means) would clarify.
3. Intuition: A small toy example (2 views, 4 clusters, controlled missing) with before/after spectral plots of the resolvent filter would let readers immediately see the ‘sharp rational filter’ effect.
4. Minor: Report per-run standard deviations (or at least note they are small) and peak memory on NUSWIDEOBJ for completeness.
5. How does the spectral gap $\lambda_c - \lambda_{c+1}$ behave as the missing rate increases on datasets like SUNRGBD or NUSWIDEOBJ? Is there a point where the gap closes and the contraction guarantee becomes vacuous?

**Strengths And Weaknesses:**

1. The spectral perspective is fresh. Most IMVC work still treats missing views as a data-completion task; reframing them as structural disconnection and attacking the problem with a resolvent filter is a clean conceptual advance. Proposition 1  and Theorem 3 are particularly convincing and directly explain why the method survives high missing rates where others collapse.

2. The dynamic feature-mapping trick that turns kernel k-means into explicit Euclidean k-means in an augmented space is elegant for large scale IMVC.

3. The paper tackles a major bottleneck in multi-view learning: scaling robust clustering to large, severely incomplete datasets. Achieving true linear complexity while fully utilizing high-order manifold structures is a significant advancement for large-scale optimization, bridging the gap between graph learning theory and scalable clustering practice.

Some concerns are as follows.

1. The paper argues that 'first-order linear fusion acts as a weak low-pass filter' and that spectral distortion is the 'core challenge'. However, the empirical comparison does not include a carefully designed first-order baseline within the same anchor-based family. To fully substantiate the claim, I would have liked to see a direct comparison to a 'first-order' version of CARD across all settings.

2. The Advantage at low missing rates is modest. The paper correctly emphasizes strength under severe incompleteness, but ICML readers will wonder whether the method is simply “very good when things are broken” rather than strictly superior in the full spectrum. A brief discussion of why the gap narrows when ρ is small (e.g., conservative α choice) would help.

3. The manuscript is dense. Moving the derivation of the contraction factor into the main text and leaving only proof sketches in the appendix would improve readability.

---

> ### Author Rebuttal · Authors · 2026-03-30
>
> W1：We implemented an apples-to-apples first-order anchor baseline with the same anchor construction and the same official graph hyperparameters as CARD; the only change is that the baseline uses the first-order operator $A = QQ^\top$ and performs K-means on $Q$.
> We focus on the severe-missingness regime, where the theoretical advantage should matter most. The results support CARD clearly. In terms of ACC, on SUNRGBD, the first-order baseline obtains 0.1682 and 0.1538 at $\rho=0.7$ and $\rho=0.9$, while CARD achieves 0.1870 and 0.1624. On NUSWIDEOBJ, the first-order baseline obtains 0.1202 and 0.1202, while CARD achieves 0.1560 and 0.1542. The same comparison is also consistently in favor of CARD in NMI and ARI. This indicates that, under severe incompleteness, the gain is not from anchorization alone, but from the high-order resolvent design.
>
> W2：Under mild missingness, the raw anchor spectrum is already relatively clean, so there is less structural distortion for the resolvent to correct. When $p$ is small, first-order smoothing is often already sufficient; the benefit of high-order diffusion becomes more visible as missingness increases.
>
> W3: We will move the definition and interpretation of the contraction factor $\nu(\alpha)$ into the main text and keep only proof sketches in the appendix.
>
> Q1：We additionally visualize ACC versus missing rate and $\alpha$; the figure is available at: [alpha_missing_NUSWIDEOBJ.pdf](https://anonymous.4open.science/r/CARD-8CB1/Rebuttal/alpha_missing_ACC_NUSWIDEOBJ_5v_30000n_31c_fea_zscore_m10_k5_bar3.pdf). The main pattern is that performance changes much more with the missing rate than with $\alpha$: for each fixed missing ratio, ACC remains relatively flat across the tested $\alpha$ values. This suggests that the near-optimal $\alpha$ range is broad and does not shift noticeably with missingness, so a fixed moderate $\alpha$ already works reasonably well in practice.
>
> Q2：We believe the scenario described in the review is substantially more severe than the setting considered in our paper. In our experiments, $\rho$ follows the standard IMVC protocol, where each sample retains at least one observed view; the case where many samples effectively have only one observed view is therefore much more extreme than our $\rho=0.9$ setting and is fundamentally topological rather than merely statistical. To address the anchor-quality concern directly, we compared K-means and random anchors under severe missingness. K-means is consistently better: on SUNRGBD, ACC drops from 0.1870 / 0.1624 to 0.1338 / 0.1294 at $\rho=0.7/0.9$; on NUSWIDEOBJ, it drops from 0.1560 / 0.1542 to 0.1159 / 0.1291. This suggests that anchor quality matters, but observed-data K-means anchors remain a fair and stable choice in the standard incomplete-IMVC regime.
>
> Q3：We have added a toy spectral visualization; see [toy_spectral_visualization](https://anonymous.4open.science/r/CARD-8CB1/Rebuttal/toy_spectral_visualization.png). In the lower-right panel, the x-axis is the mode index (spectral modes sorted from large to small), and the dashed vertical line marks the cluster boundary. The resolvent preserves the leading cluster-related modes while suppressing residual modes more aggressively: in the toy example, the gap after the 4th mode increases from 0.3290 to 0.8711.
>
> Q4：We now report seed-level standard deviations on a fixed evaluation split. For NUSWIDEOBJ, we reran CARD 10 times on the same cached splitat ρ = 0.5 / 0.7 / 0.9 using the best cached hyperparameters for each missing ratio; the ACC standard deviations are 0.0043 / 0.0081 / 0.0120, respectively. We also measured process-level peak memory on NUSWIDEOBJ by polling the MATLAB process every 200 ms; across these runs, the maximum observed peak working set and peak private memory are 10.78 GB and 28.21 GB, respectively.
>
> Q5:We have added a real-data spectral-gap plot on SUNRGBD and NUSWIDEOBJ; see [spectral_gap_visualization](https://anonymous.4open.science/r/CARD-8CB1/Rebuttal/spectral_gap_visualization.png). The empirical gap $\lambda_c-\lambda_{c+1}$ remains nonzero across the tested missing ratios: about $2.1\times10^{-3}$ to $3.7\times10^{-3}$ on SUNRGBD, and $9.3\times10^{-4}$ to $8.8\times10^{-3}$ on NUSWIDEOBJ. Thus, we do not observe the gap closing, so the contraction guarantee does not become vacuous.

---

> > ### Author Rebuttal · Reviewer_Lgt8 · 2026-04-02
> >
> > Thank you for your detailed rebuttal. My concerns are adequately addressed. I maintain my score and it could be accepted.

---

### Decision · Program_Chairs · 2026-04-30

**Decision:**

Accept (regular)

**Comment:**

The reviewers are broadly consistent in recommending acceptance. In my reading, the paper has its strengths in fresh reformulation of incomplete multi-view clustering. Meanwhile, it has technically solid high-order resolvent diffusion design with meaningful theoretical support, and the strong empirical performance together with good scalability. The rebuttal also addressed the main questions satisfactorily. Overall, I recommend acceptance.